# Mixture-of-Transformers: A Sparse and Scalable Architecture for Multi-Modal Foundation Models

**Weixin Liang\***                                                                                                    *wxliang@stanford.edu*
*Department of Computer Science*
*Stanford University*

**Lili Yu[†], Liang Luo[†], Srinivasan Iyer, Ning Dong, Chunting Zhou**
**Gargi Ghosh, Mike Lewis, Wen-tau Yih, Luke Zettlemoyer**
**Xi Victoria Lin**                                                                                                   *victorialin@meta.com*
*Meta AI*

**Reviewed on OpenReview:** *https://openreview.net/forum?id=Nu6N69i8SB*

## Abstract

The development of large language models (LLMs) has expanded to multi-modal systems capable of processing text, images, and speech within a unified framework. Training these models demands significantly larger datasets and computational resources compared to text-only LLMs. To address the scaling challenges, we introduce Mixture-of-Transformers (MoT), a sparse multi-modal transformer architecture that significantly reduces pretraining computational costs. MoT decouples non-embedding parameters of the model by modality—including feed-forward networks, attention matrices, and layer normalization—enabling modality-specific processing with global self-attention over the full input sequence. We evaluate MoT across multiple settings and model scales. In the Chameleon 7B setting (autoregressive text-and-image generation), MoT matches the dense baseline's performance using only 55.8% of the FLOPs. When extended to include speech, MoT reaches speech performance comparable to the dense baseline with only 37.2% of the FLOPs. In the Transfusion setting, where text and image are trained with different objectives, a 7B MoT model matches the image modality performance of the dense baseline with one third of the FLOPs, and a 760M MoT model outperforms a 1.4B dense baseline across key image generation metrics. System profiling further highlights MoT's practical benefits, achieving dense baseline image quality in 47.2% of the wall-clock time and text quality in 75.6% of the wall-clock time (measured on AWS p4de.24xlarge instances with NVIDIA A100 GPUs).

---

[*]Work done during a Meta internship
[†]Joint second authorship

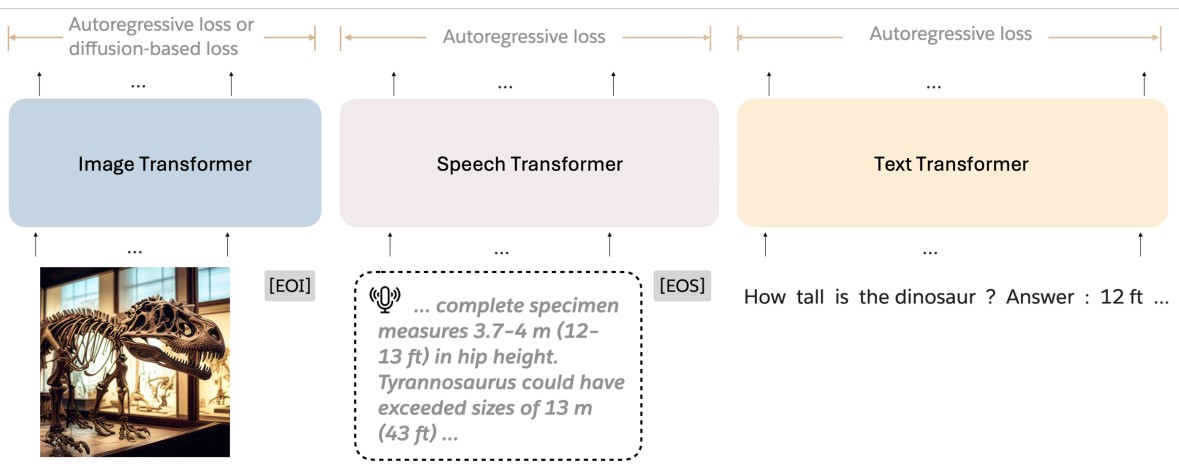

Figure 1: **Mixture-of-transformer (MoT) architecture.** MoT is a generative model architecture designed to process sequences consisting of arbitrarily interleaved modalities (e.g. text, image and speech). Each modality employs a separate set of non-embedding transformer parameters – including the feedforward network, attention matrices and layer normalization, while global self-attention is applied to the full sequence. During training, each modality can be supervised using modality-specific losses.

# 1   Introduction

The development of foundation models has expanded to multi-modal large language models (LLMs) capable of processing diverse data types—such as text, images, and speech—within a unified framework. Recent advancements, such as Chameleon (Chameleon Team, 2024), demonstrate the potential of early-fusion, mixed-modal models to generate diverse media types within a single architecture. These models hold promise for advancing applications such as content creation and cross-modal translation but pose significant computational challenges due to the complexity of simultaneously learning representations across multiple modalities.

Training early-fusion multi-modal LLMs demands significantly larger datasets and computational resources compared to single-modality models. For example, Chameleon (Chameleon Team, 2024) is trained on 9.2 trillion training tokens (including image tokens) to match LLaMA2 (Touvron et al., 2023b), which is trained on 2 trillion training tokens for text performance. Each modality introduces unique optimization challenges, which must be addressed concurrently within a unified model. Empirically, these modalities often exhibit conflicting training dynamics in a dense transformer model (Figure 15), complicating optimization and increasing computational load. Despite processing inputs as uniform tokens without modality-specific priors, different modalities occupy distinct regions in the feature space (Figure 2, Appendix Figure 23), indicating the inherent differences in how modalities are processed.

To address this scaling challenge, a promising approach is model sparsity, such as Mixture of Experts (MoE), which enables scaling by activating only a subset of model components for each input, reducing the overall computational load. In MoE, a learned router in each transformer layer sparsely activates one of multiple MLPs, allowing different experts to focus on different aspects of the data (Jacobs et al., 1991; Eigen et al., 2013; Shazeer et al., 2017; Lepikhin et al., 2020; Fedus et al., 2022; Jiang et al., 2024; Sukhbaatar et al., 2024). However, MoE introduces a number of challenges: the learned router often results in imbalanced expert utilization, requiring additional load-balancing techniques during training. Furthermore, the bi-level optimization nature of MoE complicates training dynamics, which can become unstable as model sizes scale up. Addressing these challenges in MoE remains an open area of research.

In multi-modal contexts, previous work (Bao et al., 2022b; Wang et al., 2022; Shen et al., 2023; Lin et al., 2024) has introduced *modality-aware sparsity* in the MoE layers of transformers, or further fine-tuned modality-specific modules on LLM backbones during post-training (Wang et al., 2023; He et al., 2024). These approaches have shown promising results, suggesting that a simple rule-based routing by modality outperforms the learned routing commonly used in MoE. This success might be attributed to more stable training dynamics, avoiding the instability that arises when both experts and routers are under-trained in the early stages.

Inspired by these insights, we propose **Mixture-of-Transformers (MoT)**, a sparse multi-modal transformer architecture that introduces *modality-aware sparsity* for all non-embedding transformer parameters (Figure 2a). Different from previous approaches, MoT applies *modality-aware sparsity* across the entire transformer, rather than specific layers or modules. MoT takes an interleaved multi-modal sequence (e.g., text, image, speech) as input and dynamically applies distinct, modality-specific parameters to each token, including feed-forward networks, attention projection matrices, and layer normalization. Therefore, the MoT design yields a sparse model with the exact same computational structure and FLOP count as its dense transformer counterpart.

We evaluated MoT by pretraining thirteen instances, including three 7B models, from scratch across various multi-modal settings. This comprehensive setup allowed us to assess MoT's performance in multiple experimental configurations, each progressively introducing more complex training objectives and modalities. Specifically, we conducted experiments on the following multi-modal scenarios to evaluate MoT's adaptability and efficiency gains:

1. **Autoregressive objectives for both text and images (*Chameleon*).** In the *Chameleon* setting (Chameleon Team, 2024), our 7B MoT matched the performance of a 7B dense baseline while using only 55.8% of the FLOPs as evaluated on multiple data distribution (Figure 5). Results are consistent across multiple other model scales (37M, 94M, 443M, 1.5B) (Figure 6, Appendix Figure 24).

2. **Introducing speech as a third modality (*Chameleon: Text+Image+Speech*).** When extended to include discrete speech tokens as the third modality in the Chameleon setting, MoT achieves similar performance across all modalities, with even fewer (37.2%) training FLOPs required for the speech modality (Figure 8). Results are also consistent across multiple other model scales (Figure 8, Appendix Figure 25).

3. **Autoregressive objectives for text and diffusion-based objectives for images (Transfusion).** In the *Transfusion* setting, where text and image are trained with different objectives—autoregressive for text but diffusion-based for images—our 760M MoT model, which utilizes half the training/inference FLOPs of the 1.4B dense baseline (Transfusion), outperforms the dense model across multiple metrics, including CLIP score and FID score for image generation, CIDEr score for image captioning, and image modality training loss (Figure 11). A 7B MoT model matches the image performance of the dense baseline with less than one third of the FLOPs on diffusion validation loss for image generation and CIDEr score for image captioning (Figure 10). Additionally, across three different model scales (163M, 760M, 1.4B) in the Transfusion setting, MoT consistently achieves substantial speedup in the image modality, outperforming the dense model by a wide margin (Figure 12).

To provide a deeper and more comprehensive evaluation of MoT, we extended our analysis with additional experiments to validate MoT's advantages across multiple dimensions. These experiments assessed MoT's computational efficiency, reductions in wall-clock time, and effectiveness relative to other sparse architectures:

4. **Wall-Clock Time Comparison** Furthermore, system profiling (on AWS p4de.24xlarge instances with NVIDIA A100 Tensor Core GPUs) demonstrated that MoT's efficiency translates into significant reductions in wall-clock training time. Our 7B MoT matches the image performance of the 7B dense model in just 47.2% of the time, and the text performance in 75.6% of the time (Figure 19).

5. **Comparing MoT against Mixture-of-Experts** To validate that MoT's observed gains are not merely due to additional sparse parameters (although these additional sparse parameters do not increase the training/inference FLOPs), we incorporated a 4-expert mixture-of-expert model (MoE-4x) as additional baseline throughout the experiments. MoE-4x, which includes more sparse parameters than MoT across all experiment settings, consistently underperformed compared to MoT especially in non-text modality (image, speech). The advantage of MoT over MoE-4x is even larger when measured in wall-clock time (Figure 19).

6. **Combining the Best of Both Worlds—Mixing Heterogeneous Transformers** As an early proof of concept, we explored a hybrid approach that integrates sparse transformers in the MoT framework. Specifically, we adopt the MoE-4x architecture for the text transformer of MoT, while preserving the original MoT architecture for image tasks. Preliminary results validate that this combination can further enhance text modality performance in both the Chameleon and Transfusion settings without compromising image generation quality (Figure 16, Figure 17).

## 2 Method: Mixture-of-Transformers Architecture

### 2.1 Background: Foundation Models for Multi-Modal Generation

Recent advances in large language models have expanded to modalities beyond text. A key approach tokenizes non-text data such as images and speech into discrete token sequences, and applies auto-regressive sequence modeling to the data similar to text-based models (Figure 2a). For example, Chameleon (Chameleon Team, 2024) tokenizes images into 1,024 discrete tokens using a pre-trained image tokenizer (Gafni et al., 2022) allowing unified training across text and images. Similar methods have been applied to speech (Nguyen et al., 2024). Alternative approaches like Transfusion (Zhou et al., 2024) use continuous image tokens and diffusion-based training objective to improve generation of continuous modalities such as image (Section 3.4).

To probe the internal representations of multi-modal foundation models, we analyzed their feature space. Results reveal clustering by modality (text, speech, image) across layers (Figure 2b, Appendix Figure 23).

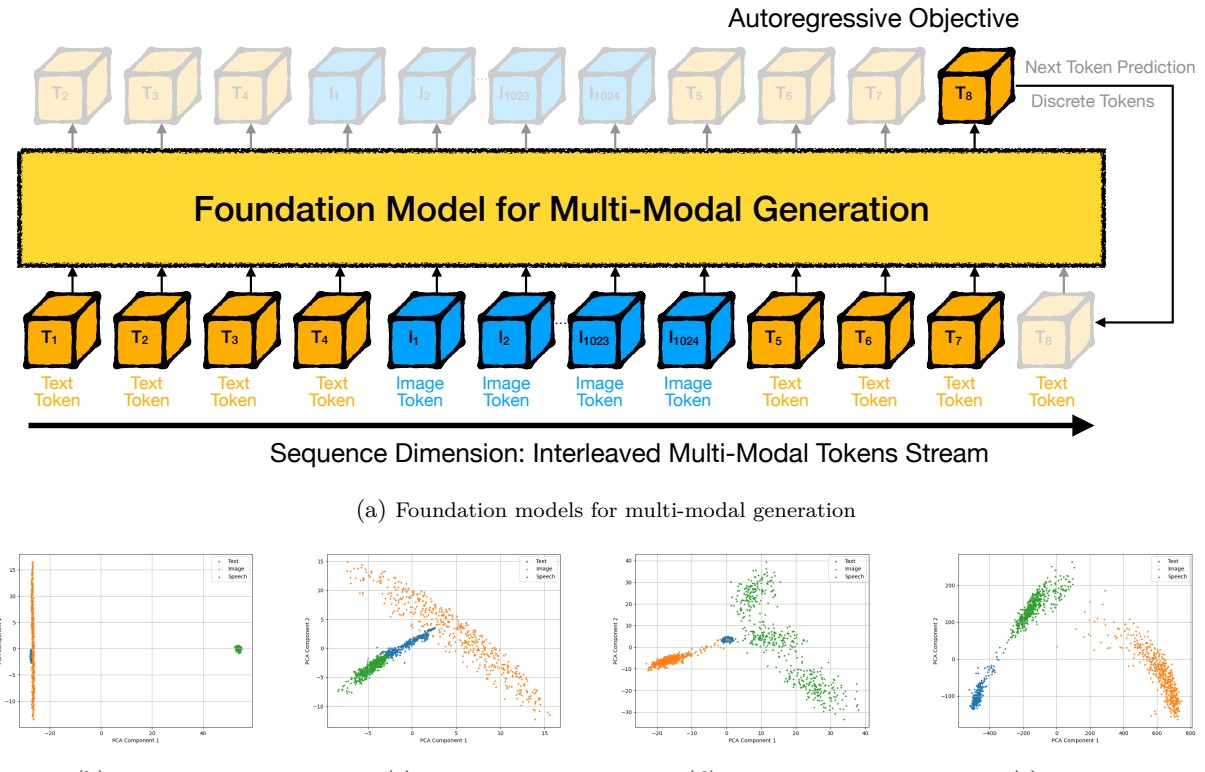

(a) Foundation models for multi-modal generation

(b) Layer 1      (c) Layer 5      (d) Layer 17      (e) Layer 32

Figure 2: **Multi-modal foundation model architecture and feature space analysis. a**, Typical multi-modal foundation model processing interleaved text (T) and image (I) tokens (e.g., *Chameleon* (Chameleon Team, 2024)). Image tokens are derived from a pre-trained VQGAN model, converting an image into 1,024 discrete tokens. **b**, Principal Component Analysis of latent feature space for *Chameleon+Speech* 7B Dense model across layers 1, 5, 17, and 32.[†] Despite the model's architecture processing all inputs as uniform discrete tokens without modality-specific priors, distinct clustering by modality (text, speech, image) is observed in the feature space. This natural clustering highlights the inherent differences between modalities, suggesting that the model might have processed them differently.

Principal component analysis (PCA) shows distinct regions for different modalities in the feature space, despite uniform processing of inputs as discrete tokens without modality-specific priors. This natural clustering suggests inherent differences in modality processing, informing our subsequent approach.

## 2.2 Mixture-of-Transformers Architecture: Modality-Specific Parameter Decoupling

We present Mixture-of-Transformers (MoT), a novel architecture designed to accelerate multi-modal pretraining while reducing computational costs. MoT extends the standard transformer architecture by incorporating modality-specific weights for all non-embedding model parameters, including feed-forward networks, attention matrices, and layer normalization. This approach allows the model to process different modalities more efficiently while preserving the ability to learn cross-modal interactions. Let $x = (x_1, \ldots, x_n)$ be the input sequence of tokens, where each $x_i$ belongs to a modality $m_i \in \{\text{text}, \text{image}, \text{speech}\}$. A typical transformer layer can be expressed as:

$$
\begin{aligned}
a &= \text{Attn}(x, \theta_{\text{attn}}) \\
h &= x + \text{LayerNorm}_{\text{attn}}(a) \\
\text{output} &= h + \text{LayerNorm}_{\text{ffn}}(\text{FFN}(h, \theta_{\text{ffn}}))
\end{aligned}
\tag{1}
$$

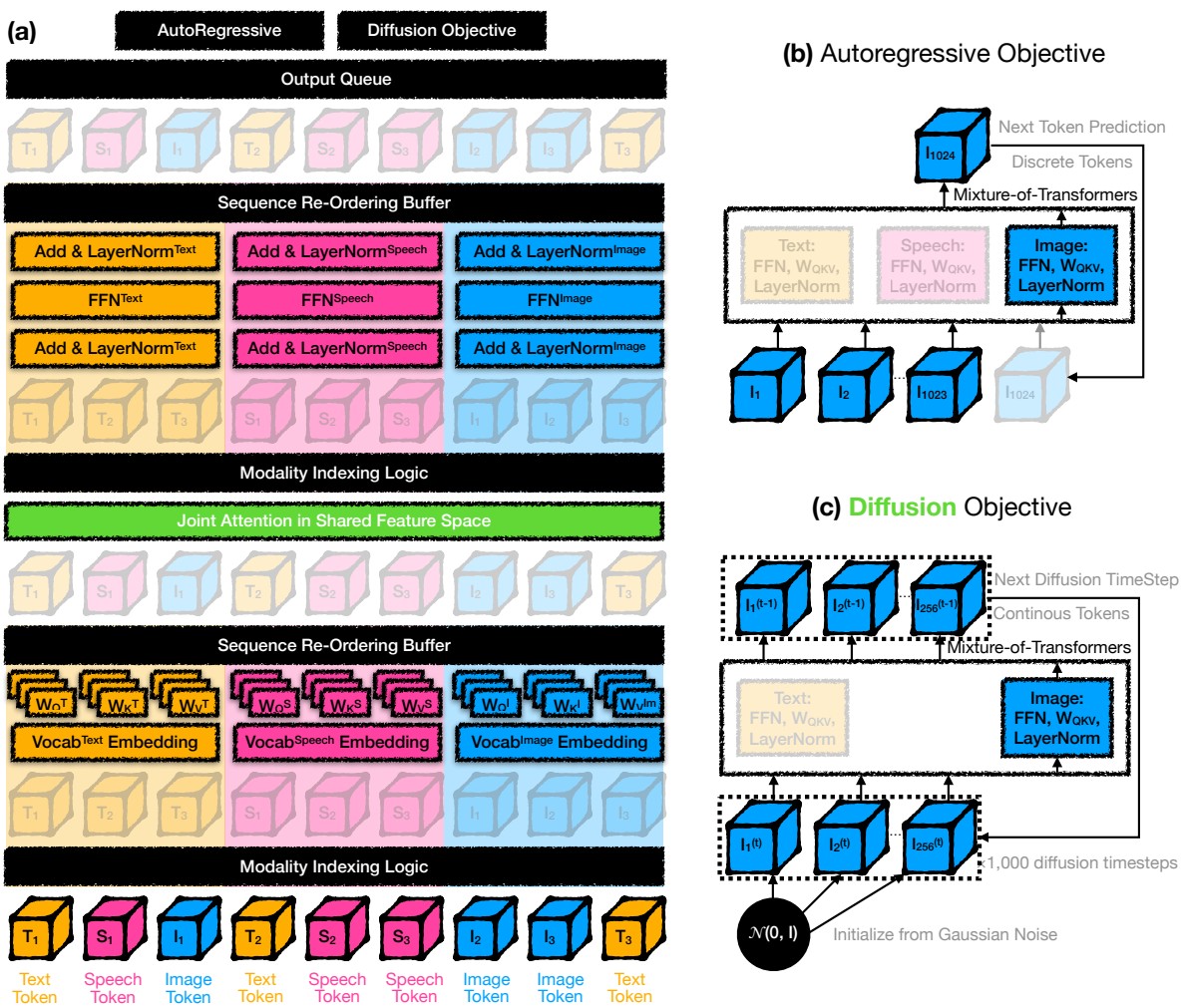

Figure 3: **Mixture-of-Transformers architecture for multi-modal generative AI. a**: Schematic of the sparsely activated Mixture-of-Transformers (MoT) architecture. For each input token, MoT activates modality-specific weights (including feed-forward networks, attention projection matrices, and layer normalization), then applies self-attention across the entire sequence. $T$, $S$, and $I$ indicate text, speech, and image tokens, respectively. **b-c**: Flexibility in modality representation and training objectives. Images can be represented as **(b)** a sequence of discrete tokens trained with an autoregressive objective (Chameleon setting) or **(c)** a sequence of continuous tokens trained with a diffusion objective (Transfusion setting). This allows integration of diverse learning tasks, such as autoregressive objectives for text and diffusion-based objectives for images.

---

**Algorithm 1** Mixture-of-Transformers (MoT) Computation

---

1: Let $x = (x_1, \ldots, x_n)$ be the input sequence, where $x_i \in \mathbb{R}^d$ and $m_i \in \{text, image, speech\}$ is the modality of $x_i$
2: Let $\mathcal{M} = \{text, image, speech\}$ be the set of modalities
3: **for** each modality $m \in \mathcal{M}$ **do**
4:       $I_m \leftarrow \{i : m_i = m\}$                                                                    ▷ Indices of tokens for modality $m$
5:       $X_m \leftarrow \{x_i : i \in I_m\}$                                                                    ▷ Group tokens by modality
6:       $Q_m \leftarrow W_Q^m X_m, K_m \leftarrow W_K^m X_m, V_m \leftarrow W_V^m X_m$                         ▷ Modality-specific projections
7: **end for**
8: $Q \leftarrow \bigcup_{m \in \mathcal{M}} Q_m, K \leftarrow \bigcup_{m \in \mathcal{M}} K_m, V \leftarrow \bigcup_{m \in \mathcal{M}} V_m$                    ▷ Restore original sequence order
9: $A \leftarrow \text{softmax}\left(\frac{QK^T}{\sqrt{d_k}}\right) V$                                                    ▷ Global self-attention
10: **for** each modality $m \in \mathcal{M}$ **do**
11:       $O_m \leftarrow W_O^m A_{I_m}$                                                                    ▷ Modality-specific output projection
12:       $H_m \leftarrow X_m + \text{LayerNorm}_{\text{attn}}^m(O_m)$                                        ▷ Residual connection and layer norm
13:       $F_m \leftarrow \text{FFN}_m(H_m)$                                                                ▷ Modality-specific feed-forward network
14:       $Y_m \leftarrow H_m + \text{LayerNorm}_{\text{ffn}}^m(F_m)$                                        ▷ Residual connection and layer norm
15: **end for**
16: **return** $\{Y_m : m \in \mathcal{M}\}$                                                                ▷ Return transformer layer outputs

---

In our proposed MoT, we decouple the parameters by modality while maintaining global self-attention[1]:

$$a = \text{GlobalAttn}(x, \{\theta_{\text{attn}}^m\}_{m \in \{\text{text,image,speech}\}})$$
$$h_i = x_i + \text{LayerNorm}_{\text{attn}}^{m_i}(a_i) \tag{2}$$
$$\text{output}_i = h_i + \text{LayerNorm}_{\text{ffn}}^{m_i}(\text{FFN}(h_i, \theta_{\text{ffn}}^{m_i}))$$

The global self-attention mechanism operates across all modalities, capturing cross-modal relationships despite the modality-specific parameter decoupling:

$$\text{GlobalAttn}(x, \{\theta_{\text{attn}}^m\}) = \left(\text{softmax}\left(\frac{QK^T}{\sqrt{d_k}}\right) V\right) W_O^{m_i}$$
$$Q_i = x_i W_Q^{m_i}, \quad K_i = x_i W_K^{m_i}, \quad V_i = x_i W_V^{m_i} \tag{3}$$

Here, $W_Q^{m_i}$, $W_K^{m_i}$, $W_V^{m_i}$, and $W_O^{m_i}$ are modality-specific projection matrices, and $\text{LayerNorm}_{\text{attn}}^{m_i}$ and $\text{LayerNorm}_{\text{ffn}}^{m_i}$ are modality-specific layer normalization.

This approach allows MoT to adapt its processing to the specific characteristics of each modality while maintaining a unified architecture for multi-modal learning. The computation process in MoT begins with grouping input tokens by modality (Algorithm 1, lines 3-5). Modality-specific projections are then applied for attention (line 6), followed by global self-attention across all modalities (lines 8-9). Subsequently, modality-specific output projections (line 11), layer normalization, and feed-forward networks are applied (lines 12-13). The process concludes with the combination of outputs, incorporating residual connections and layer normalization (lines 14-16).

---

[1]Comparing to works that utilize cross-attention to fuse information from different modalities (Alayrac et al., 2022; Aiello et al., 2023), our formulation using global self-attention normalizes attention weights across tokens of different modalities while reducing the number of layers in the architecture.

# 3 Experiments

## 3.1 Results Overview

We evaluated the Mixture-of-Transformers (MoT) architecture across three multi-modal experiment settings, each progressively incorporating more complex training objectives and modalities. For each setting, we compared MoT against two baselines: a dense transformer model and a Mixture-of-Experts model with 4 experts (MoE-4x). All model implementations, built upon the dense model, maintain identical FLOPs for both training and testing, enabling direct efficiency and performance comparisons.

1. **Multi-modal experiment setting with autoregressive objectives (*Chameleon*, Figure 4).** Both modalities trained using autoregressive objectives. Images represented as 1,024 discrete tokens via a pre-trained VQ-VAE model (Gafni et al., 2022). We compared MoT's performance to baselines across training and evaluation metrics for both modalities.

2. **Extended multi-modal experiment with speech modality (*Chameleon: Text+Image+Speech*, Figure 7).** Extended the previous setting by incorporating speech as a third modality, represented by discrete tokens via a pre-trained speech tokenizer. All modalities are trained with autoregressive objectives. This setting evaluated MoT's ability to handle an additional modality while maintaining efficiency and performance.

3. **Multi-modal experiment with modality-specific objectives (*Transfusion*, Figure 10)** Explored multi-objective training with text using autoregressive objectives and images using diffusion-based objectives. This experiment highlighted MoT's capacity to manage distinct training objectives for different modalities, potentially improving image generation quality while maintaining text generation capabilities.

The following sections present detailed results for each setting: Chameleon (Section 3.2), Chameleon+Speech (Section 3.3), and Transfusion (Section 3.4). Each section provides comprehensive comparisons of MoT against the baselines across various multi-modal generative evaluation metrics. In Section 3.5, we report an ablation study demonstrating the impact of model performance when introducing modality-specific decoupling to different components of a transformer.

Text: Autoregressive Objective

Image: Autoregressive Objective

Figure 4: **Multi-modal experiment setting with autoregressive objectives (*Chameleon*).** Both text and images are trained using autoregressive objectives. Images are tokenized into 1,024 discrete tokens using a pre-trained VQ-VAE model. This setting demonstrates unified processing across modalities with a single objective function.

## 3.2 Performance in the *Chameleon* Setting: Autoregressive Objectives for Text and Image Generation

In this subsection, we evaluated the Mixture-of-Transformers (MoT) architecture in the Chameleon setting, where text and image modalities are trained using autoregressive objectives.

### 3.2.1 Experiment Setup

**Data and Pre-processing.** We use the same mixed-modal training data and the same text and pre-trained image tokenizers as Chameleon Team (2024). The training data comprises roughly equal amount of text and image tokens. We evaluated the 7B model performance using validation losses on held-out sets of the Obelisc (Laurençon et al., 2023), MS-COCO (Lin et al., 2014), Flickr30k (Plummer et al., 2015), and Shutterstock[2] datasets. More specifically, for MS-COCO and Flickr30k, we take the Karpathy test split of MS-COCO (Lin et al., 2014) and the Karpathy test split of Flickr30k (Plummer et al., 2015), and report text-to-image and image-to-text conditional perplexity using these two datasets.

**Model Hyperparameters.** We evaluated MoT across multiple model scales ranging from 37M to 7B parameters, comparing it to dense transformer and MoE-4x baselines. All models were pre-trained from scratch with controlled FLOPs for fair comparison. Table 1 details the architectural specifications and training configurations for each model scale. Model architectures were scaled progressively, with hidden dimensions increasing from 256 to 4096, and layer counts from 4 to 32. Attention heads scaled from 8 to 32, while sequence length remained constant at 4096 tokens across all scales. As model size increases, we reduce batch sizes per GPU from 12 to 2, while increasing the number of GPUs from 32 to 384. Training steps were set at 160,000 for smaller models (37M to 443M) and 120,000 for larger models (880M to 7B). Total training tokens ranged from 0.168 to 0.377 trillion, with most configurations processing approximately 0.252 trillion tokens. This allowed us to examine MoT's performance across a wide range of model scales and training FLOPs, providing insights into its effectiveness at different computational scales.[3]

**Mixture-of-Experts Implementation.** For our MoE-4x baselines, we employed Expert Choice (EC) (Zhou et al., 2022) routing, a state-of-the-art routing method that ensures balanced load during training by having each expert select top-$k$ inputs based on routing weights. However, EC cannot be directly applied to auto-regressive generation, as it violates the causal dependency between tokens in a sequence, where each token is generated based solely on the previous ones. Previous work have proposed various inference-time adjustment to ensure generation causality for MoE models trained with EC routing. For example, some recent works have explored using expert choice routers out-of-the-box as token choice routers during

---

[2]https://www.shutterstock.com/

[3]With this setup, we focus on evaluating the relative performance of the proposed architecture and the baseline at various FLOPs budgets, rather than conducting a scaling law study.

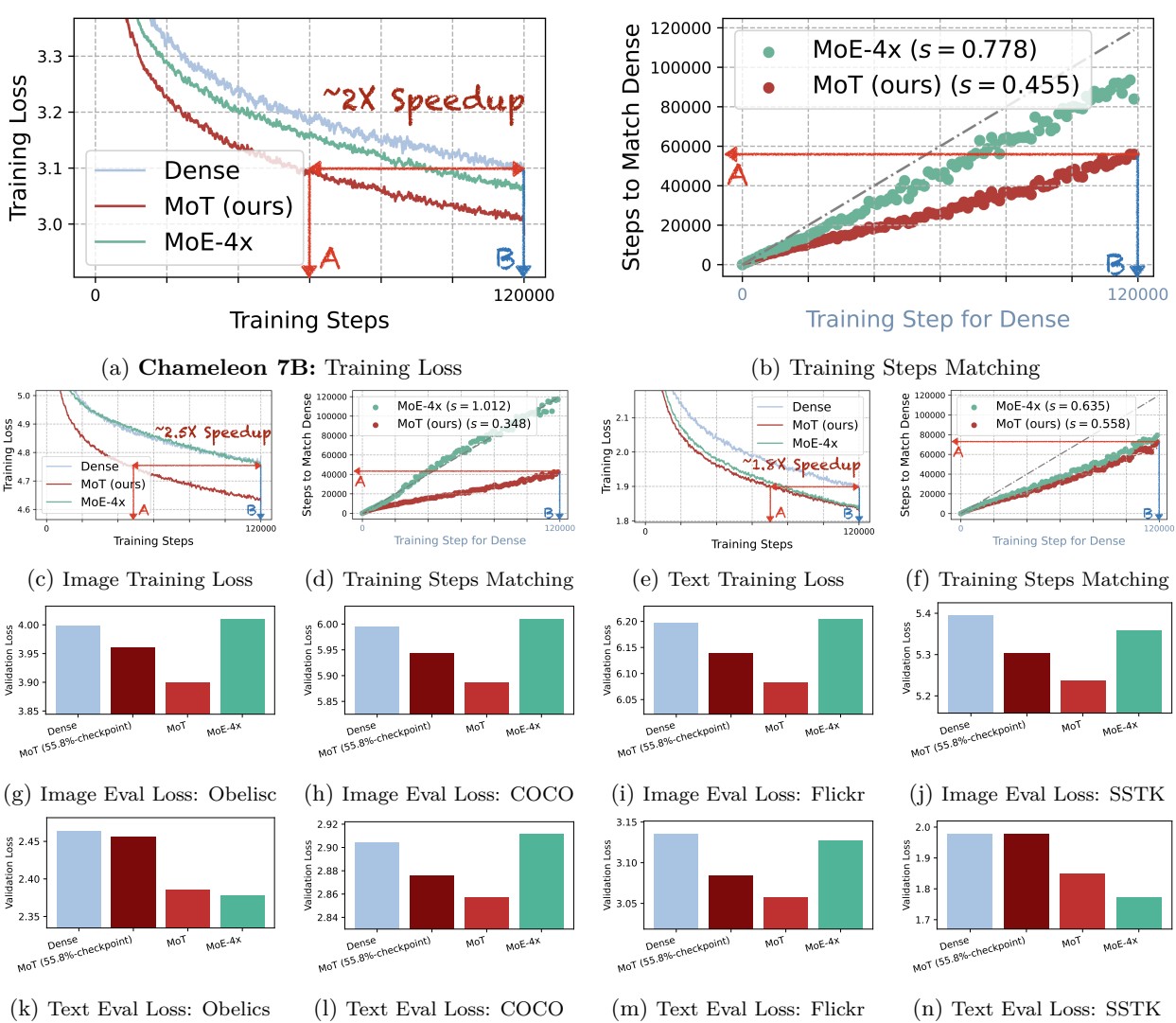

Figure 5: **Pre-training acceleration of MoT for 7B Chameleon multi-modal model. a**, Global training loss curves. MoT reduces loss faster than dense and MoE-4x models, matching dense model's final loss at 120,000 steps in 60,000 steps. **b**, Step matching plot for training loss in a. MoT requires 45.5% of dense model's training steps for comparable performance. **c,d**, Image modality training loss and corresponding step matching plot. **e,f**, Text modality training loss and corresponding step matching plot. MoT particularly effective for image modality, requiring 34.8% of dense model's training steps to match final loss. Both MoT and MoE-4x outperform dense model for text modality. **g-j**, Image modality validation losses. **k-n**, Text modality validation losses. Comparison of final validation losses for all models and MoT at 55.8% training checkpoint. MoT at 55.8% training steps achieves comparable or lower validation losses than dense model's final loss, indicating 44.2% reduction in required training FLOPs. Model sizes for sparse models indicate activated parameters. All runs are FLOPs-controlled and pre-trained from scratch.

inference (Zhong et al., 2024), or training small auxiliary MLP predictors post-training for routing (Raposo et al., 2024; Lin et al., 2024).

We evaluated all models using the same EC routing as during training, focusing exclusively on validation perplexity. This approach guarantees an isoFLOP inference setting as the dense baseline. However, it also introduces two confounding factors. First, it may overestimate MoE-4x's validation performance, as the router can access future tokens, potentially leading to information leakage. Second, it may also underestimate

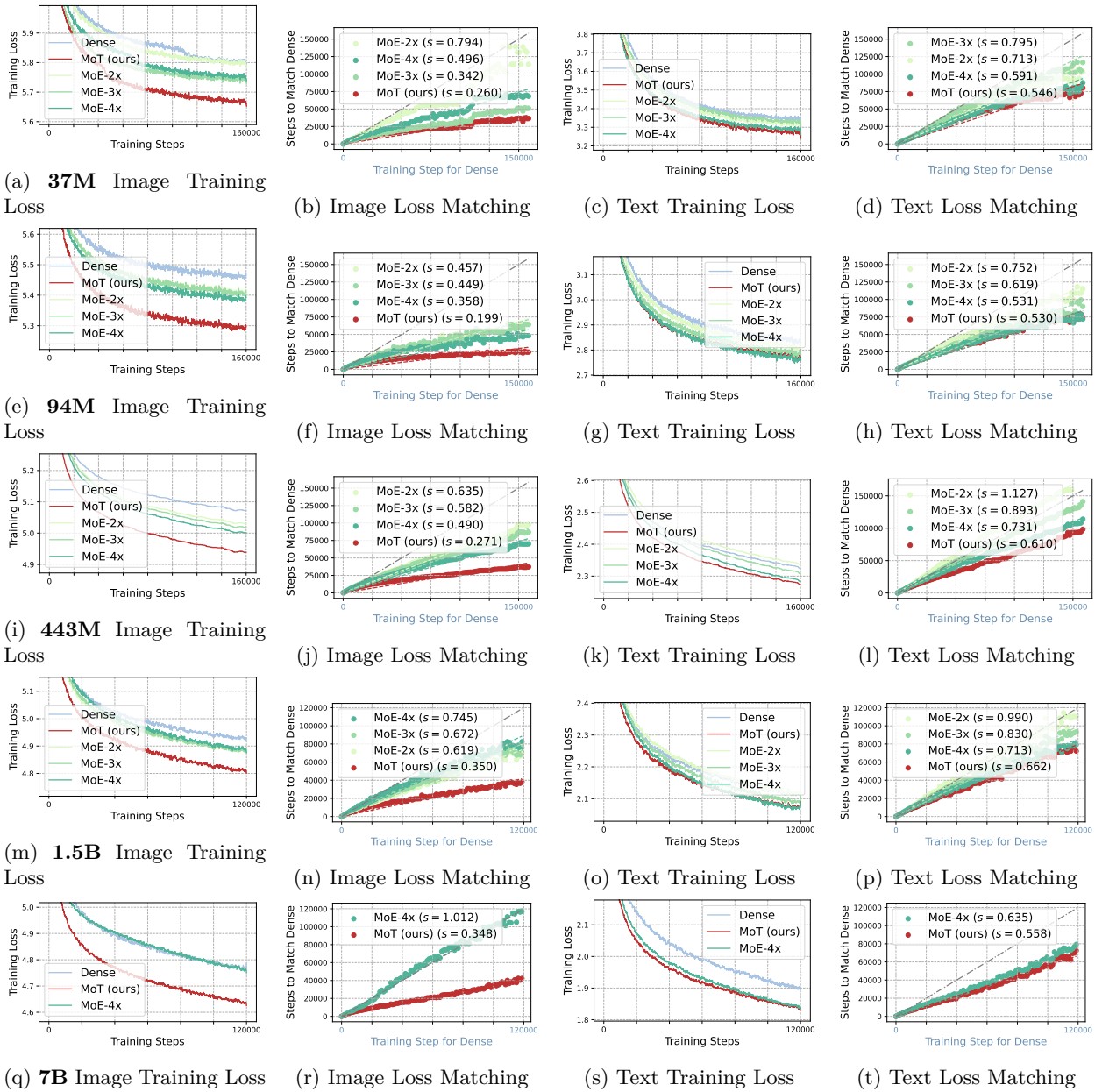

Figure 6: **Modality-specific pre-training loss and step matching plots across model scales (*Chameleon* setting).** MoT shows consistent, significant speedup in image modality across all scales (37M, 94M, 443M, 1.5B, 7B), outperforming dense and MoE-4x models. MoE-4x exhibits diminishing gains in image modality as scale increases, with advantages disappearing at 7B. In text modality, both MoT and MoE-4x outperform dense model, with MoT showing comparable or slightly better gains. Validation loss results in Appendix Figure 24). Model sizes for sparse models indicate activated parameters. All runs are FLOPs-controlled and pre-trained from scratch.

MoE-4x's validation performance when the evaluation data distribution differs significantly from the training data, resulting in uneven token distribution among experts. We acknowledge these limitations and provide additional discussion on the results compared to MoE-4x in each individual experiment to provide a more comprehensive understanding.

| Model Size | Hidden Dim. | Layers | Heads | Seq. Length | Batch Size/GPU | GPUs | Tokens/Batch | Steps | Tokens (T) |
|---|---|---|---|---|---|---|---|---|---|
| 37M | 256 | 4 | 8 | 4096 | 12 | 32 | 1.57M | 160k | 0.252 |
| 94M | 512 | 8 | 8 | 4096 | 8 | 32 | 1.05M | 160k | 0.168 |
| 443M | 1024 | 24 | 16 | 4096 | 6 | 64 | 1.57M | 160k | 0.252 |
| 1.5B | 2048 | 24 | 16 | 4096 | 4 | 128 | 2.10M | 120k | 0.252 |
| 7B | 4096 | 32 | 32 | 4096 | 2 | 384 | 3.15M | 120k | 0.377 |

Table 1: **Architectural specifications and training configurations of models across different parameter scales (Chameleon setting).** The table lists the hidden dimension, number of transformer layers, attention heads, and sequence length for each model size. Additionally, we provide the batch size used per GPU, the total number of GPUs, training steps, and the corresponding total number of training tokens (in trillions).

### 3.2.2 Accelerated Pre-Training at 7B Scale

The Mixture-of-Transformers (MoT) architecture demonstrated significant pre-training acceleration at the 7B parameter scale (Figure 5a). MoT achieved the dense model's final loss (at 120k steps) in half the time, reaching equivalent performance at just 60k steps. We quantified this acceleration using step matching analysis (Figure 5b). This method plots the training steps required by MoT and MoE-4x to reach equivalent loss values as the dense model. The analysis revealed that MoT consistently required only 45.5% of the dense model's training steps to achieve comparable pre-training loss, indicating a substantial and sustained acceleration throughout training. Modality-specific analysis showed MoT's particular effectiveness in the image modality, requiring only 34.8% of the dense model's training steps to match final loss (Figure 5c-f). MoE-4x showed limited improvement in this domain. For text, both MoT and MoE-4x outperformed the dense model, with MoT showing comparable or slightly better gains.

Validation loss results (Figure 5g-n) further supported these findings. MoT at 55.8% of training steps achieved validation losses comparable to or lower than the dense model's final validation loss across both modalities. This indicates that MoT requires only 55.8% of the training FLOPs to match the dense model's validation metrics, offering substantial computational savings.

### 3.2.3 Performance Across Multiple Model Scales

We extended our analysis of MoT to five additional model scales (37M, 94M, 443M, 1.5B, and 7B) within the Chameleon setting (Figure 6). MoT consistently delivered significant speedups in the image modality across all scales, outperforming both the dense model and MoE-4x.

We also observed that MoE-4x exhibited diminishing returns as model size increased. While it showed some speedup in image modality at smaller scales, this advantage diminished at the 7B scale. In contrast, MoT maintained its performance edge across all scales. For text modality, both MoT and MoE-4x outperformed the dense model, with MoT showing comparable or slightly better gains.

Validation loss results across these scales (Appendix Figure 24) remains consistent with the trend observed in training loss. In both image and text modalities, MoT consistently achieved lower validation losses with fewer training steps compared to the dense model and MoE-4x. This demonstrates that MoT's benefits extend across a wide range of model sizes, highlighting its versatility and efficiency for large-scale multi-modal generative tasks.

### 3.3 Extending to a Third Modality: Chameleon Text+Image+Speech Results

We evaluated MoT's performance in a multi-modal setting by introducing speech as a third modality along-side text and images. Experiments focused on the 7B model and smaller scales, comparing MoT against dense and MoE-4x models pre-trained from scratch under FLOPs-controlled conditions.

#### 3.3.1 Experiment Setup

We utilized the training dataset from SpiRit-LM (Nguyen et al., 2024) (Table 2) as our speech dataset. The training data included both speech-only samples and interleaved speech/text data in the SpiRit-LM format. Speech input was converted to discrete tokens using an in-house tokenizer, a variant of DinoSR (Liu et al., 2024a), which extracts semantic tokens with a vocabulary size of 500. Each token represents 40ms of audio content (25Hz). Model architectural specifications and training configurations of models are shown in Table 3.

To create the three-modality training dataset, we combine the speech training dataset with the Chameleon text-and-image training dataset with a sampling ratio of 1:6. Within each dataset, we adopt the same data mix ratio as utilized by Nguyen et al. (2024) and Chameleon Team (2024).

This experimental setup allows us to evaluate MoT's capacity to handle complex multi-modal inputs, including the temporal and semantic challenges inherent in speech processing, while maintaining efficiency and performance across text, image, and speech modalities. We followed the evaluation setup in the aforementioned Chameleon setting (Section 3.2.1) and additionally reported the speech modality validation losses on held-out sets of LibriLight (LL60K) and People's Speech Dataset (PPL30K).

| Dataset | Modality | Hours | # Speech Tokens[†] | # Text Tokens |
|---|---|---|---|---|
| People's Speech Dataset (Galvez et al., 2021) | Speech-only | 16,404 | 1.2B | – |
| Voxpopuli (English) (Wang et al., 2021) | Speech-only | 23,166 | 1.6B | – |
| LibriLight (Kahn et al., 2020) | Speech-only | 55,308 | 4B | – |
| Multilingual LibriSpeech (English) (Pratap et al., 2020) | Speech+Text | 44,585 | 3.2B | 0.5B |
| Spotify (Clifton et al., 2020) | Speech+Text | 57,290 | 4.2B | 0.7B |

Table 2: Dataset information for speech pre-training. [†]The speech token counts are computed after deduplication.

| Model Size | Hidden Dim. | Layers | Heads | Seq. Length | Batch Size/GPU | GPUs | Tokens/Batch | Steps | Tokens (T) |
|---|---|---|---|---|---|---|---|---|---|
| 443M | 1024 | 24 | 16 | 4096 | 6 | 64 | 1.57M | 160k | 0.252 |
| 880M | 1536 | 24 | 24 | 4096 | 4 | 128 | 2.10M | 120k | 0.252 |
| 1.5B | 2048 | 24 | 16 | 4096 | 4 | 128 | 2.10M | 120k | 0.252 |
| 7B | 4096 | 32 | 32 | 4096 | 2 | 384 | 3.15M | 120k | 0.377 |

Table 3: **Architectural specifications and training configurations of models across different parameter scales (Chameleon+Speech setting).** The table lists the hidden dimension, number of transformer layers, attention heads, and sequence length for each model size. Additionally, we provide the batch size used per GPU, the total number of GPUs, training steps, and the corresponding total number of training tokens (in trillions).

#### 3.3.2 Performance with Speech Integration at 7B Scale

The 7B MoT model with added speech modality (Figure 8) demonstrates substantial pre-training acceleration. In the speech modality, MoT speeds up pre-training substantially compared to the dense and MoE-4x models (Figure 8a). Step matching analysis (Figure 8b) shows MoT achieving equivalent speech pre-training loss to the dense model in just 22.9% of the training steps, indicating considerable computational efficiency.

MoT also consistently outperforms baselines according to the validation loss results on speech datasets LL60K and PPL30K (Figure 8c-f). Notably, MoT maintained its efficiency across image and text modalities

Figure 7: **Extended multi-modal experiment with speech modality (*Chameleon+Speech*).** Building on the previous setting, a third modality (speech) is incorporated. All three modalities—text, images, and speech—are trained using autoregressive objectives. Speech is represented as discrete tokens via a pre-trained speech tokenizer, showcasing the model's ability to handle diverse input types uniformly.

(Figure 8g-n), achieving comparable or lower validation losses than the dense model's final loss at only 55.8% of training steps. This demonstrates MoT's robust performance in multi-modal settings.

### 3.3.3 Scalability Across Model Sizes

We extended our evaluation to smaller model scales (443M, 880M, 1.5B) in the *Chameleon: Text+Image+Speech* setting (Figure 9). MoT consistently delivered significant acceleration across all three modalities, with pronounced improvement in speech processing. MoT required only 15.1% to 33.6% of the dense model's training steps to match speech training loss across all scales.

MoE-4x exhibited inferior performance in speech tasks, showing improvements in training loss but unstable generalization in validation, particularly for speech (Figure 9). We observed mixed performance of MoE-4x across all scales studied (443M, 880M, 1.5B), where it underperforms the dense baseline in speech validation loss, despite outperforming it in speech pre-training loss. We believe this instability stems from two reasons: first from the learned routing mechanism (specifically, the Expert Choice routing in Section 3.2.1) which is sensitive to data distribution shifts. In heterogeneous multi-modal settings, the routing network can become imbalanced; for example, when speech tokens are less frequent or display a distinct distribution relative to text and images, the gating mechanism may under-utilize some experts. These result in suboptimal performance on the speech validation datasets, LL60K and PPK30K, which exhibit significantly different data distributions compared to the combined text and speech training data. Second, MoE-4x's large number of raw parameters could make it prone to overfitting, hence contributing to its underperformance in speech validation loss, especially given the smaller amount of unique speech tokens in the combined dataset (Section 3.3.1).

In contrast, MoT leverages a deterministic, modality-aware partitioning that decouples non-embedding parameters (such as FFNs and projection matrices) by modality. This design inherently avoids the load-balancing issues of MoE and offers a more stable signal across modalities, allowing it to consistently outperform both dense and MoE-4x models across all scales in speech modality, for both training and validation metrics. This consistency demonstrates MoT's effective adaptation to multi-modal tasks, highlighting its reliability and scalability in generative AI applications across speech, image, and text modalities.

We highlight that in our FLOP-controlled experiments, we ensured that the dense baseline and MoT models have nearly identical parameter counts in the non-embedding portions of the network. The MoT's efficiency does not come from reducing the overall number of parameters but from decoupling modality-specific components. In contrast, MoE-4x inherently has a larger parameter count due to multiple expert branches. Although this increased count might suggest a potential for improved performance, our results indicate that the extra parameters in MoE do not translate into stable or superior performance across modalities—largely because of the aforementioned routing challenges.

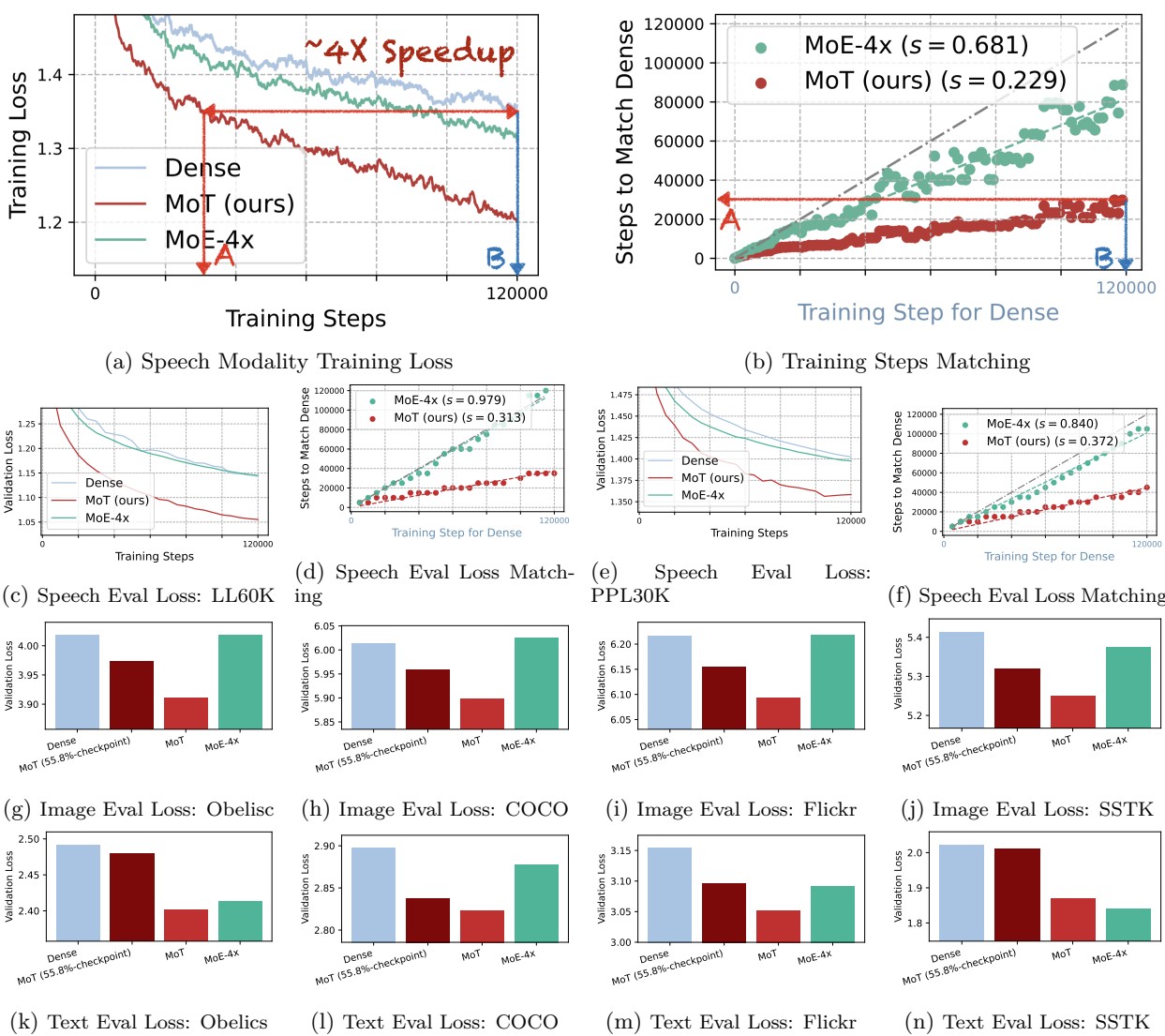

Figure 8: **Performance of MoT with speech as a third modality. a**, MoT accelerates pre-training for speech modality, reducing loss faster than dense and MoE-4x models. **b**, Step matching plot shows MoT achieves equivalent loss in 22.9% of dense model's training steps, indicating substantial computational efficiency. **c-f**, Validation losses on LL60K and PPL30K speech datasets confirm MoT's consistent performance. MoT reaches baseline speech performance in 37.2% of the FLOPs (**f**). **g-n**, MoT maintains efficiency across image and text modalities when speech is added. At 55.8% of training steps (determined from Figure 5 - Chameleon 7B), MoT achieves comparable or lower validation losses than dense model's final loss for image and text tasks. Model sizes for sparse models indicate activated parameters. All runs are FLOPs-controlled and pre-trained from scratch.

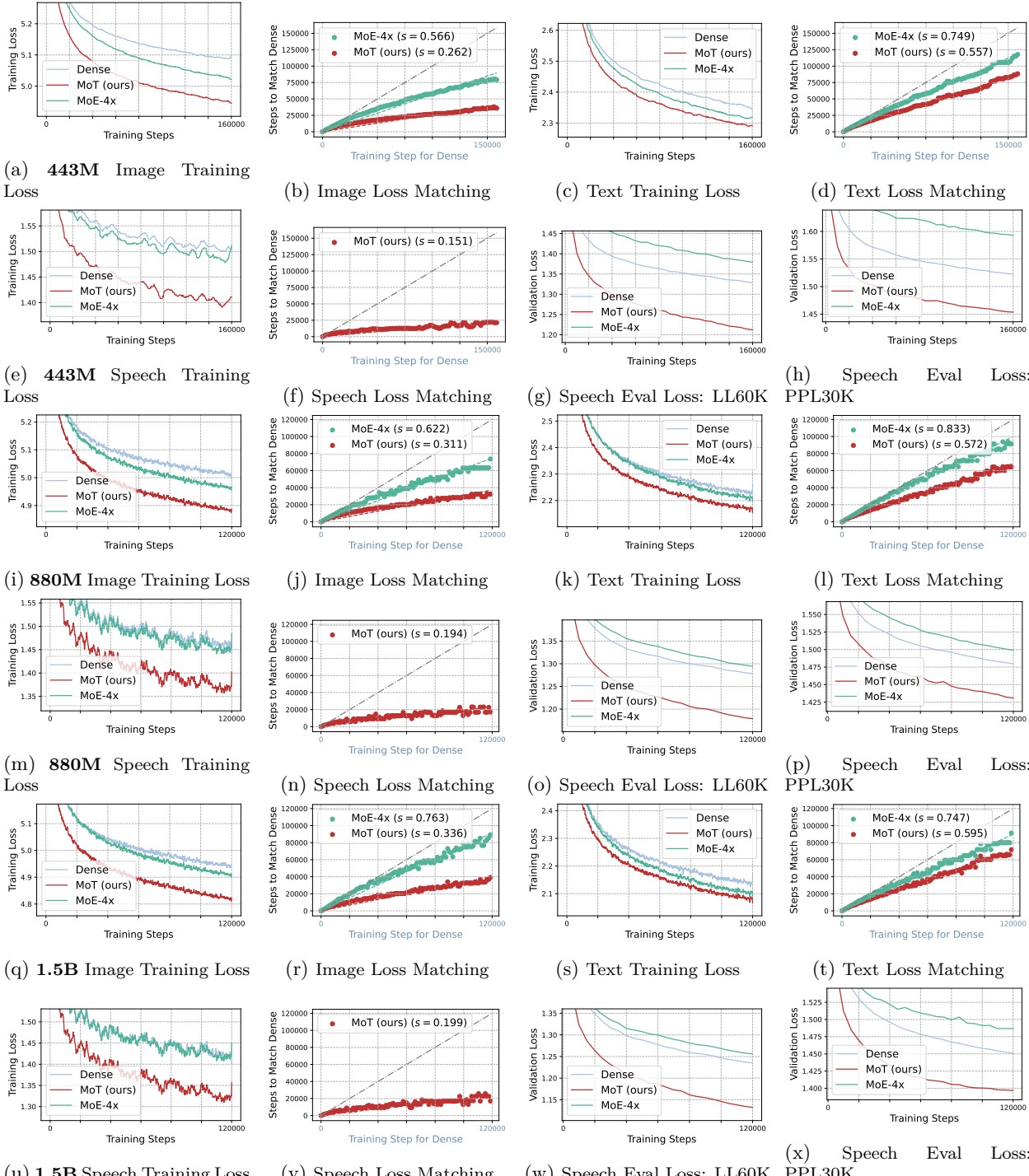

Figure 9: **Speech, image, and text modality performance across model scales.** MoT demonstrates consistent speedup across image and text modalities for models ranging from 443M to 1.5B parameters (see Appendix Figure 25 for validation losses). Speech modality shows even greater acceleration, with MoT matching dense model training loss in 15.1%-33.6% of steps across all scales. MoT also consistently outperforms MoE-4x in speech modality. Sparse model sizes indicate activated parameters. All runs are FLOPs-controlled and pre-trained from scratch.

### 3.4 Multi-Objective Training in the Transfusion Setting: Autoregressive Text and Diffusion-Based Image Generation

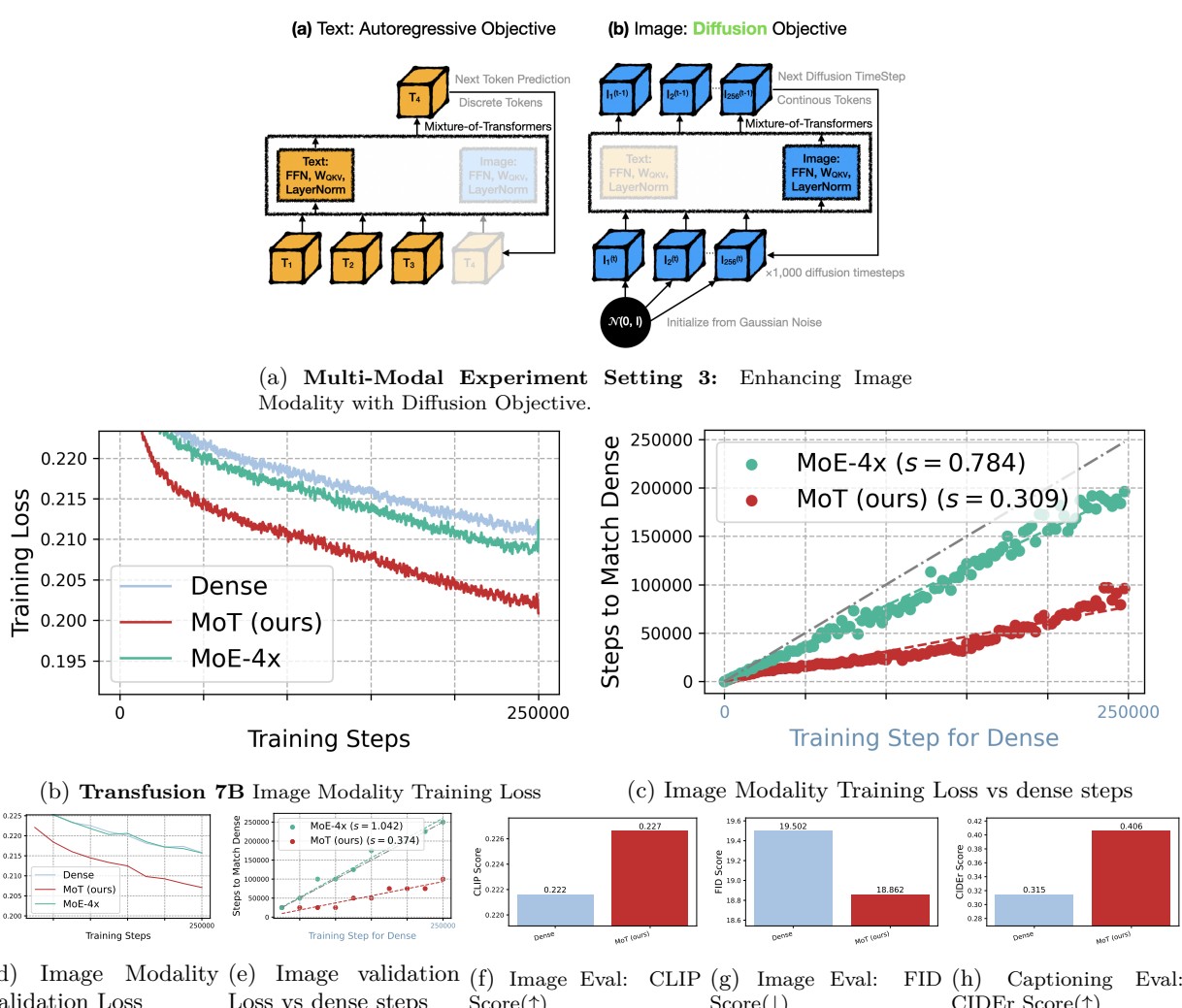

(a) **Multi-Modal Experiment Setting 3:** Enhancing Image Modality with Diffusion Objective.

(b) **Transfusion 7B** Image Modality Training Loss

(c) Image Modality Training Loss vs dense steps

(d) Image Modality validation Loss

(e) Image validation Loss vs dense steps

(f) Image Eval: CLIP Score(↑)

(g) Image Eval: FID Score(↓)

(h) Captioning Eval: CIDEr Score(↑)

Figure 10: **Multi-objective training performance in MoT. a**, Schematic of multi-objective setup: text trained with autoregressive objectives, images with diffusion-based objectives, as described in *Transfusion* (Zhou et al., 2024). **b-e**, MoT accelerates pre-training beyond Transfusion, particularly for image modality. The 760M MoT model, using half the training/inference FLOPs of the 1.4B dense baseline (Transfusion), consistently outperforms the dense model across metrics: CLIP score (0.214 vs 0.206, higher is better), FID score (21.145 vs 24.688, lower is better), CIDEr score for image captioning (0.320 vs 0.286, higher is better), and image modality training loss. All runs are FLOPs-controlled and pre-trained from scratch, demonstrating MoT's superior efficiency and performance across various model scales.

Transfusion (Zhou et al., 2024) introduces a unified framework that enables a single transformer model to process both discrete text and continuous image modalities (Appendix A). The key innovation is the utilization of separate loss functions for each modality—language modeling loss for text and diffusion loss for images —while sharing data and parameters within a single architecture. In this subsection, we evaluate the performance of MoT under the multi-objective training setup in the Transfusion setting. Here, text is trained using autoregressive objectives, while images are trained using diffusion-based objectives. All models are pre-trained from scratch under FLOPs-controlled conditions.

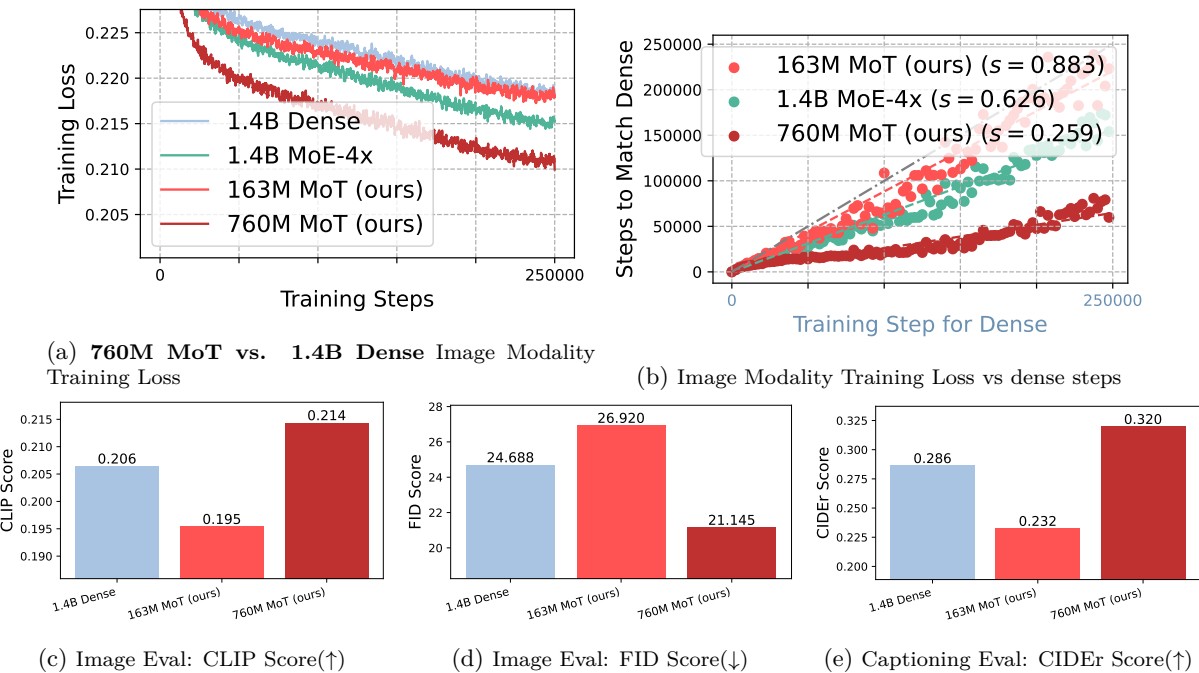

(a) **760M MoT vs. 1.4B Dense** Image Modality Training Loss

(b) Image Modality Training Loss vs dense steps

(c) Image Eval: CLIP Score(↑)

(d) Image Eval: FID Score(↓)

(e) Captioning Eval: CIDEr Score(↑)

Figure 11: **In the Transfusion setting, a 760M MoT model outperforms a 1.4B dense baseline across key image generation metrics, while using only half the FLOPs for both training and inference. a-b**, MoT accelerates pre-training beyond Transfusion, particularly for image modality. The 760M MoT model, using half the training/inference FLOPs of the 1.4B dense baseline (Transfusion), consistently outperforms the dense model across metrics: CLIP score (0.214 vs 0.206, higher is better, (**c**)), FID score (21.145 vs 24.688, lower is better, (**d**)), CIDEr score for image captioning (0.320 vs 0.286, higher is better, (**e**)), and image modality training loss. All runs are FLOPs-controlled and pre-trained from scratch, demonstrating MoT's superior efficiency and performance across various model scales.

### 3.4.1 Experiment Setup

**Data and Pre-processing.** We adopt the same data setup as Zhou et al. (2024). For text, we utilize the Llama 2 tokenizer and corpus (Touvron et al., 2023b), which contains 2 trillion tokens across diverse domains. Images are encoded into latent patches using a Variational Autoencoder (VAE) (Kingma & Welling, 2022), where each patch corresponds to a continuous vector. We use a collection of 380 million licensed Shutterstock images and captions. Each image is center-cropped and resized to $256 \times 256$ pixels. Our VAE model does $8 \times 8$ spatial downsampling of the image. For multimodal examples, we enclose each image sequence with special tokens—*beginning of image* (BOI) and *end of image* (EOI)—before integrating it into the text sequence. This approach results in a single sequence that may contain both discrete elements (text tokens) and continuous elements (image patches). We randomly order the images and captions, placing the caption first 80% of the time. In most of our experiments, we sample 0.5 trillion tokens (or patches) from two modalities at a 1:1 ratio.

**Model Hyperparameters.** To investigate scaling trends, we train models at five different sizes – 0.16B, 0.76B, 1.4B, and 7B parameters. Model architectural specifications and training configurations of models are shown in Table 4. We keep U-Net patch encoding parameters fixed as 0.27B additional parameters across all configurations. We randomly initialize all model parameters, and optimize them using AdamW ($\beta_1 =0.9$, $\beta_2 =0.95$, $\epsilon =$1e-8) with a learning rate of 3e-4, warmed up for 4000 steps and decaying to 1.5e-5 using a cosine scheduler. We train on sequences of 4096 tokens in batches of 2M tokens for 250k steps, reaching 0.5T tokens in total. We regularize with weight decay of 0.1 and clip gradients by norm (1.0). We conduct 250 diffusion steps during inference.

**Evaluation Benchmarks.** We evaluate the model's performance on a collection of standard unimodal and cross-modal benchmarks. For text-to-text tasks, we measure perplexity on 20 million held-out tokens from Wikipedia and the C4 corpus (Raffel et al., 2020). For text-to-image tasks, we report the diffusion validation loss[4] following SD 3 (Esser et al., 2024) on held-out Conceptual 12M (CC12M; Changpinyo et al. (2021)) data. We also use the MS-COCO benchmark (Lin et al., 2014), where we generate images based on 30,000 randomly selected prompts from the validation set. We measure the photorealism of these images using zero-shot Fréchet Inception Distance (FID) (Heusel et al., 2017) and their alignment with the prompts using CLIP score (Radford et al., 2021).[5] We also assess the model's ability to generate image captions by reporting CIDEr (Vedantam et al., 2015) scores on the Karpathy test split of MS-COCO (Lin et al., 2014).

| Model Size | Hidden Dim. | Layers | Heads | Seq. Length | Batch Size/GPU | GPUs | Tokens/Batch | Steps | Tokens (T) |
|---|---|---|---|---|---|---|---|---|---|
| 163M | 768 | 16 | 12 | 4096 | 4 | 128 | 2.10M | 250k | 0.524 |
| 760M | 1536 | 24 | 24 | 4096 | 4 | 128 | 2.10M | 250k | 0.524 |
| 1.4B | 2048 | 24 | 16 | 4096 | 2 | 256 | 2.10M | 250k | 0.524 |
| 7B | 4096 | 32 | 32 | 4096 | 2 | 256 | 2.10N | 250k | 0.524 |

Table 4: **Architectural specifications and training configurations of models across different parameter scales (Transfusion setting).** The table lists the hidden dimension, number of transformer layers, attention heads, and sequence length for each model size. Additionally, we provide the batch size used per GPU, the total number of GPUs, training steps, and the corresponding total number of training tokens (in trillions).

### 3.4.2 Mixture of Transformers Enhances Multi-Objective Training Efficiency

In the Transfusion setting (Figure 10), the Mixture-of-Transformers (MoT) architecture demonstrates significant acceleration in pre-training for the image modality at the 7B parameter scale (Figure 10b). Compared to dense and MoE-4x models (Figure 10c), MoT substantially speeds up pre-training. Step matching analysis indicates that MoT achieves equivalent image pre-training loss to the dense model in only 30% of the training steps. MoT's efficiency in image generation is further evidenced by superior diffusion validation loss (Figure 10d,e), higher COCO-30k CLIP scores (Figure 10f), and lower FID scores (Figure 10g). To compare our 7B model with external models, we compute the COCO-30k FID at a guidance level of 1.6 and obtain a score of 8.14. In contrast, a dense model trained on 1T tokens with richer data achieves a COCO-30k FID of 9.22 under the same guidance level. This comparison further validates the efficiency of MoT over dense models. In image understanding tasks, MoT exhibits more than a threefold speedup compared to the dense model, achieving a final score of 40.6 versus 31.5. We exclude the MoE-4x caption performance due to potential information leakage from expert choice training. These results extend our findings from the Chameleon 7B setting, where MoT matched the dense baseline's image pre-training loss using only 34.8% of the FLOPs (Figure 5).

The text performance improvement of MoT in the Transfusion setting was less pronounced compared to the Chameleon setting. In Chameleon, MoT required only 54.6% to 66.2% of training steps to match the dense model's text modality training loss. In Transfusion, the text performance improvement was marginal to none (see Appendix 26). We believe several factors contribute to this observation: (1) this discrepancy may be attributed to the separate training objectives for image and text modalities in the Transfusion setting, which leads to close to optimal text performance. The decoupling of training objectives in the dense model might confer benefits similar to MoT's decoupling of modality weights. This hypothesis is supported by observations in Zhou et al. (2024), where only changing the image training representation and objective led

---

[4]SD 3 (Esser et al., 2024) and Movie Gen (Polyak et al., 2024) show that diffusion validation loss is a strong predictor of overall model performance. Validation loss is well correlated with human evaluations of text alignment and overall quality, as well as with holistic image evaluation metrics, including GenEval (Ghosh et al., 2023) and T2I-CompBench (Huang et al., 2023).

[5]For clarity, unless otherwise noted, we do not use classifier-free guidance for image generation (Ho & Salimans, 2022) for the ease of comparisioon. Although classifier-free guidance offers immediate improvements in image quality—as indicated by better FID and CLIP scores—it requires complex hyperparameter tuning to find the optimal guidance value for each individual model, complicating the evalutation process. For 7B MOT and dense model, we use classifier-free guidance of 5 to generate example images.

to dramatic improvements in text performance compared to Chameleon (Chameleon Team, 2024), despite no direct changes to text training. (2) Since our experiments are FLOP-controlled, the limited improvement margin on text tasks could reflect the fact that text generation is already efficient and may require less model sparsity. (3) The modality decoupling in MoT is especially beneficial for modalities that demand heavier computation (like image diffusion), so the relative advantage for text—which is less compute-intensive—is smaller. None the less, we acknowledge the need for further investigation into text performance and plan to explore additional modifications or hybrid strategies (e.g., integrating MoE elements selectively) in future work.

At smaller scales, MoT demonstrated significant performance gains, particularly in the image modality (Figure 11). A 760M parameter MoT model, operating at half the FLOPs of a 1.4B parameter dense baseline, consistently outperformed its larger counterpart across multiple metrics. Image quality improved, as evidenced by CLIP (0.214 vs 0.206) and FID (21.145 vs 24.688) scores (Figure 11c,d). Image captioning capability, measured by CIDEr score, also improved (0.320 vs 0.286; Figure 11e).

When comparing models with 8-fold difference in size (163M MoT vs. 1.4B dense/MoE-4x), the 163M MoT achieves comparable image modality training loss. While the 163M MoT still slightly lags behind the 1.4B dense model in evaluation metrics, the more than 8-fold reduction in both training and inference compute highlights MoT's strength in the image modality.

### 3.4.3 Scalability Across Model Sizes

MoT consistently outperformed baselines in image modality tasks across all scales (163M, 760M, 1.4B, Figure 12). FID scores showed substantial improvements for MoT over dense models: 21.58 vs 27.42 (163M), 15.75 vs 25.58 (760M), and 15.85 vs 19.32 (1.4B). CLIP scores also consistently improved with MoT: 0.195 vs 0.185 (163M), 0.214 vs 0.202 (760M), and 0.217 vs 0.206 (1.4B).

In text modality, MoT matched dense models in training and validation loss on text-only datasets. However, MoT demonstrated significantly better generalization in captioning tasks, with consistently higher CIDEr scores: 0.232 vs 0.147 (163M), 0.320 vs 0.251 (760M), and 0.335 vs 0.286 (1.4B). As discussed in 7B results, MoT shows little improvement on text accross the scales.

### 3.4.4 Performance with fine-tuning

Following the original Transfusion (Zhou et al., 2024) setup, we fine-tune the 7B MoT and dense models on an internal visually appealing dataset and on image editing tasks, as shown in Figure 13. The fine-tuned models are capable of generating text, detailed hand features, fictional images, and photorealistic images. After fine-tuning, MoT demonstrates better quality and higher faithfulness compared to the fine-tuned dense models (see Appendix B), indicating that the performance gain of MoT over the dense baseline is well maintained after fine-tuning. Notably, we train on only 0.5 trillion tokens, which is significantly less than other state-of-the-art (SOTA) image generation models. Zhou et al. (2024) shows that the model is not yet saturated even at 2 trillion tokens; we leave the scaling up of our model and training with more data as future work. We also finetune 7B MoT on 8k image eiditing data (Zhang et al., 2023), as show in Figures 13 (i,j) . The Transfusion MoT fine-tuned on image editing tasks extends its capabilities to generate images based on other images by following text instructions.

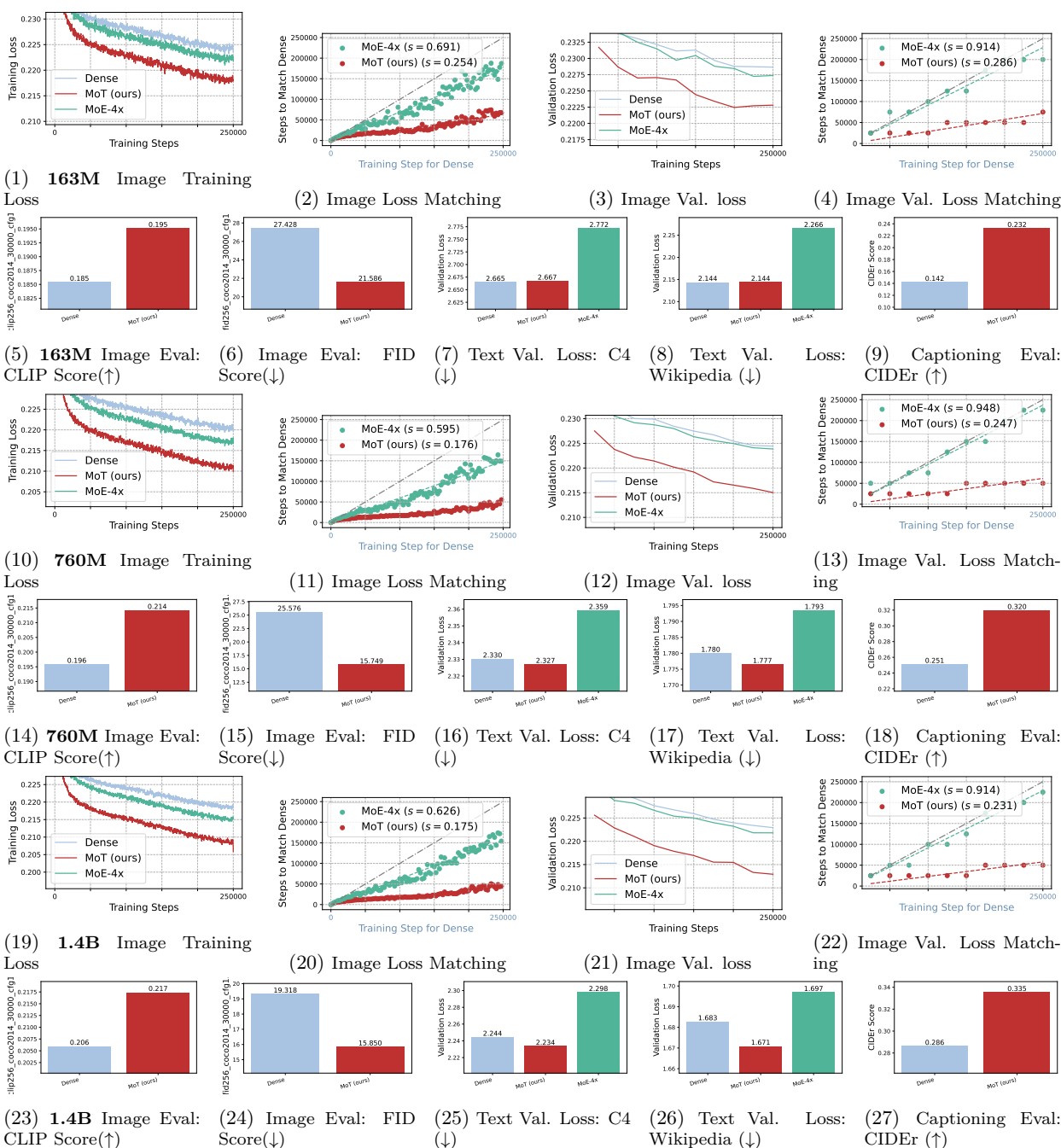

Figure 12: **Modality-specific training loss and step matching plots in Transfusion setting across model scales.** MoT consistently achieves substantial speedup in image modality (trained with diffusion-based objectives) across 163M, 760M, and 1.4B models, outperforming dense model and MoE-4x. In image modality, MoT reaches comparable training loss to dense model in 17.5%-26.4% of steps across all scales. For text modality, MoT matches dense model in training and validation loss on C4 and Wikipedia datasets, with improved generalization in captioning tasks (CIDEr score). MoE-4x shows unstable performance: lower training losses (Appendix Figure 26) but poorer generalization than dense model on text evaluation metrics. Model sizes for sparse models indicate activated parameters. All experiments FLOPs-controlled and pre-trained from scratch.

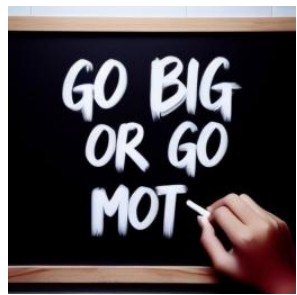

(a) "GO BIG OR GO MOT" is written on the blackboard.

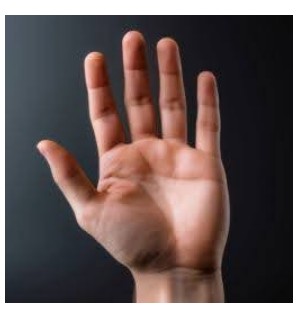

(b) A close up photo of a human hand, hand model. High quality

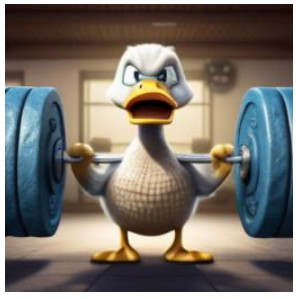

(c) An angry duck doing heavy weightlifting at the gym.

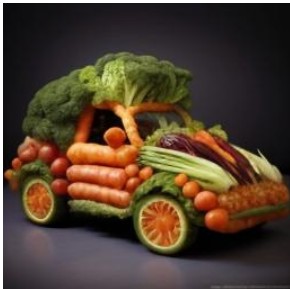

(d) A car made out of vegetables.

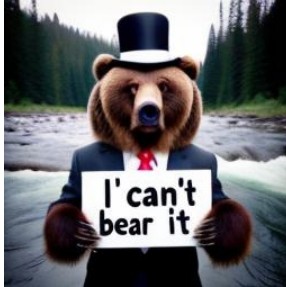

(e) photo of a bear wearing a suit and tophat in a river in the middle of a forest holding a sign that says "I cant bear it".

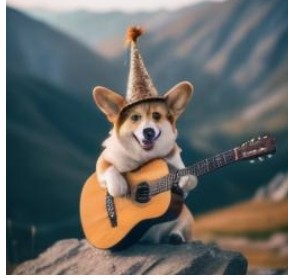

(f) A photo of a corgi dog wearing a wizard hat playing guitar on the top of a mountain.

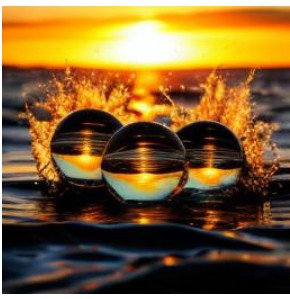

(g) Three spheres made of glass falling into ocean. Water is splashing. Sun is setting.

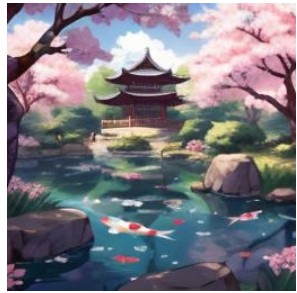

(h) A tranquil, anime-style koi pond in a serene Japanese garden, featuring blossoming cherry trees.

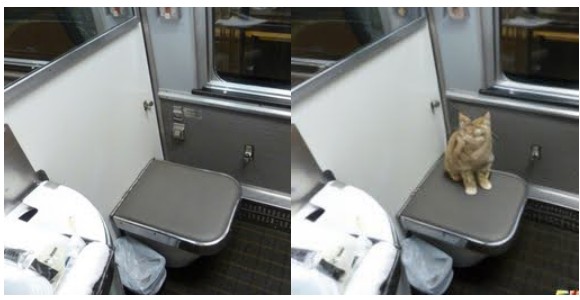

(i) Put a cat on the seat.

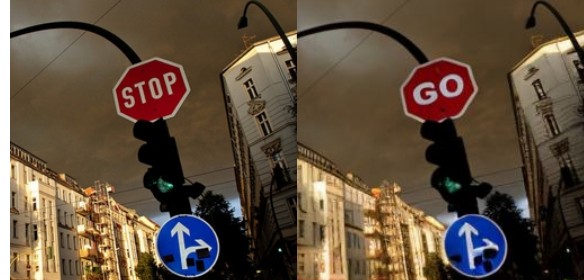

(j) Change the stop sign to say "GO".

Figure 13: Image generation and image editing (last row) examples from a 7B Transfusion MOT model trained with 0.5T tokens.

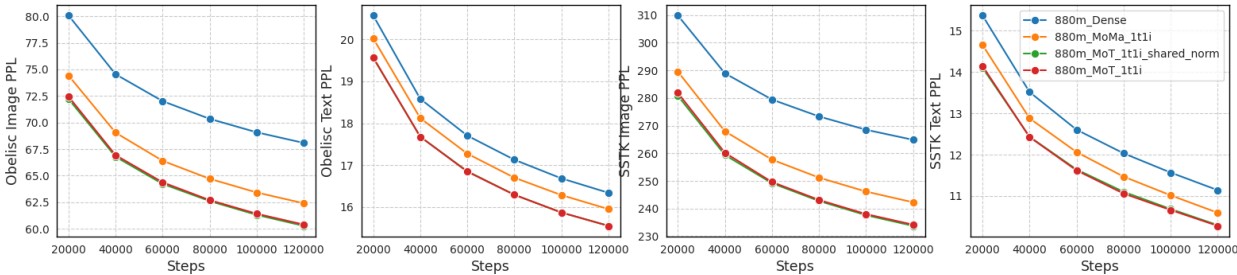

Figure 14: Ablation results of modality untying in different transformer components in the Chameleon setting, evaluated using the held-out sets of Obelisc and Shutterstock. Modality untying in the feedforward module alone significantly improves model performance, with substantial gains on the image modality. Further untying Q, K, V matrices in the attention module yields significant performance improvements, whereas untying layer norms has a negligible impact on evaluation performance.

## 3.5 Impact of Modality Untying in Different Transformer Components

We conduct ablation experiments to understand the impact of modality untying in different transformer components. We conduct the experiments using architectures with FLOPs controlled to match the dense architecture in Table 5. We compare four model variations: (1) the dense baseline; (2) a model with modality untying only in the feedforward module as in Lin et al. (2024); (3) a model with modality untying in both the feedforward module and the Q, K, V weight matrices (excluding LayerNorms); (4) the full modality-untied MoT model.

| Model Size | Hidden Dim. | Layers | Heads | Seq. Length | Batch Size/GPU | GPUs | Tokens/Batch | Steps | Tokens (T) |
|---|---|---|---|---|---|---|---|---|---|
| 880M | 1536 | 24 | 24 | 4096 | 4 | 128 | 2.10M | 120k | 0.252 |

Table 5: **Architectural specifications and training configurations of model used in ablation experiments.**

As shown in Figure 14, modality untying in the feedforward module alone (Lin et al., 2024) significantly improves model performance, with substantial gains on the image modality. Further untying the Q, K, V weight matrices in the attention module yields significance performance improves. On the Obelisc (Laurençon et al., 2023) held-out set, this leads to approximately 33.3% FLOPs saving for the image modaily and 10% FLOPs saving for the text modality compared to only performing untying in the feedforward module. Notably, the FLOPs savings from adding attention untying to feedforward untying are smaller than those from adding feedforward untying to the dense model. We attribute this to two factors: (1) the feedforward module accounts for a larger proportion of FLOPs in the transformer architecture given our context length (4096), and (2) the feedforward module serves as a memory component in transformers, where employing separate memory parameters for each modality is effective. Finally, we observe that further untying the LayerNorm parameters on top of feedforward and attention untying has a negligible impact on evaluation performance.[6]

---

[6]This finding does not suggest that untying LayerNorm parameters is entirely ineffective. Our experiments only examine its impact when combined with attention and feedforward untying. We leave the understanding of its individual effectiveness to future work.

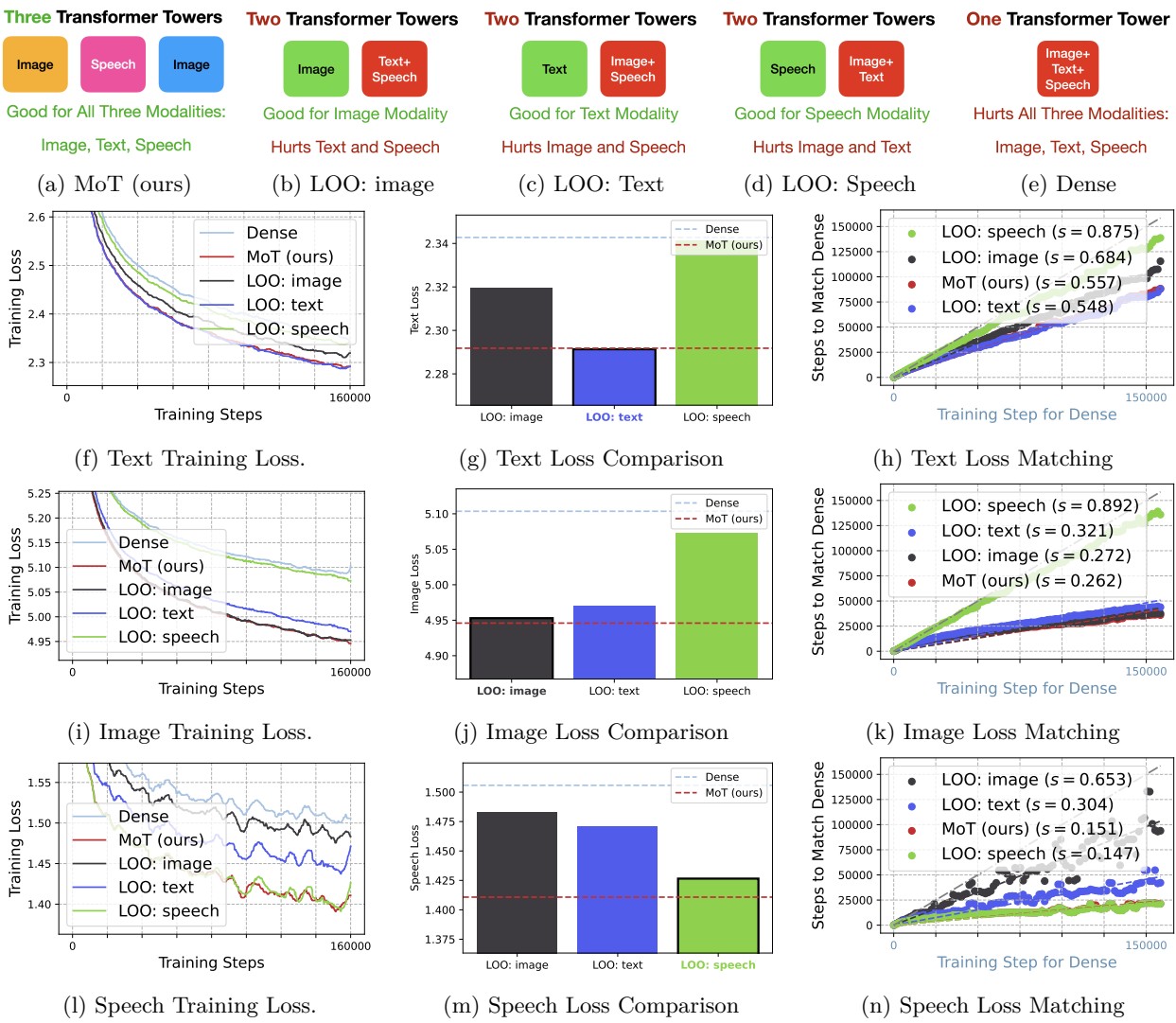

Figure 15: **Modality Leave-One-Out (LOO) analysis of MoT variants in *Chameleon+Speech* setup. a**, Proposed MoT architecture with separate transformer towers for image, text, and speech. **b-d**, Two-tower MoT variants for three modalities: **b**, Text and speech combined (LOO: image). **c**, Image and speech combined (LOO: text). **d**, Text and image combined (LOO: speech). **e**, Dense transformer with single tower for all modalities. All models (**a-e**) have equivalent FLOPs. **f-n**, Performance results across modalities. Combining modalities in a single tower consistently degrades performance, while separation improves results. LOO: text, LOO: image, and LOO: speech achieve lowest losses in their respective isolated modalities (**g,j,m**). Analysis highlights the importance of modality-specific parameter allocation for optimal performance. All models are 443M in size. For sparse models, size indicates activated parameters. All runs are FLOPs-controlled and pre-trained from scratch.

## 4 Modality Separation in MoT: Leave-One-Out Analysis

### 4.1 Experiment Setup

To evaluate the efficacy of MoT's modality-specific architecture, we conducted a Leave-One-Modality-Out (LOO) analysis with the *Chameleon: Text+Image+Speech* framework. This analysis aimed to quantify the benefits of separating modalities into distinct transformer towers, as implemented in MoT, compared to combining multiple modalities within a single tower.

Figure 15 illustrates the MoT variants and their performance across different configurations. The baseline MoT architecture (Figure 15a) comprises three separate transformers for image, text, and speech modalities. We tested three LOO variants, where two out of the three modalities share the same transformer:

- **LOO-image** (**Figure 15b**) — text and speech combined, image isolated

- **LOO-text** (**Figure 15c**) — image and speech combined, text isolated

- **LOO-speech** (**Figure 15d**) — text and image combined, speech isolated

A dense transformer architecture with all modalities in a single tower (Figure 15e) served as a baseline. All models maintained equivalent FLOPs to ensure fair comparison.

### 4.2 Results

The Leave-One-Modality-Out (LOO) analysis revealed advantages of modality separation in the MoT architecture (**Figure 15f-n**). Combining modalities consistently degraded performance, as evidenced by higher training and validation losses across configurations. The LOO-text configuration achieved the lowest text loss (**Figure 15g**), while LOO-image and LOO-speech yielded the lowest losses in their respective isolated modalities (**Figure 15j,m**). These results support the importance of modality-specific parameter allocation in MoT.

The impact of modality combination varied across configurations. In LOO-speech, separating the speech modality preserved MoT's benefits for speech but eliminated improvements in image and text modalities. Similarly, LOO-image retained most of MoT's improvements in image performance, while merging text and speech led to performance declines in both. In LOO-text, speech performance deteriorated when combined with image, yet the image modality largely maintained the gains realized by MoT. This differential impact suggests non-reciprocal modality competition effects. By dedicating separate transformer towers to each modality, MoT is able to optimize for the unique characteristics of each modality, resulting in better overall performance across all modalities.

# 5 Combining the Best of Both Worlds – Mixing Heterogeneous Transformers

## 5.1 Combining MoT and MoE-4x in the Chameleon Setting

In this subsection, we present preliminary results exploring the potential of combining key features of Mixture-of-Transformers (MoT) and Mixture-of-Experts (MoE-4x) within the Chameleon setting. Specifically, we modify the MoT architecture by incorporating MoE-4x into the text transformer tower. The text feed-forward network (FFN) of MoT is replaced with the MoE-4x mechanism, which introduces multiple expert layers to the model. The image transformer tower remains unchanged and follows the original MoT architecture. This experiment seeks to assess whether integrating MoE-4x's expertise mechanism can further enhance MoT's performance in multi-modal generative tasks.

As shown in Figure 16, the combination of MoT and MoE-4x significantly accelerates the reduction of text training loss compared to the dense model, MoE-4x, and MoT alone (**Figure 16a-b**). The results demonstrate that introducing MoE-4x into the text transformer provides additional speedup without sacrificing the efficiency gains of MoT in the image modality (**Figure 16c-d**). The averaged training losses across both text and image modalities confirm that the combined model maintains or exceeds the performance of MoT in both tasks (**Figure 16e-f**).

When evaluating the combined model's performance on validation datasets, we observe consistent gains in the text modality. As shown in **Figure 16g-j**, the combination of MoT and MoE-4x achieves the best text validation performance across multiple datasets, outperforming both MoT and MoE-4x. Importantly, the image modality performance remains comparable to or slightly better than that of MoT, indicating that the incorporation of MoE-4x into the text tower does not hinder MoT's efficiency in image generation tasks. These early results suggest that combining the strengths of MoT and MoE-4x offers a promising avenue for improving multi-modal models, particularly in tasks requiring simultaneous text and image generation.

## 5.2 Combining MoT and MoE-4x in the Transfusion Setting

We extend the experiment to the Transfusion setting, where distinct objectives are applied to different modalities—autoregressive objectives for text and diffusion-based objectives for images. Similar to the Chameleon setting, we replace the FFN layer of the text transformer in MoT with a 4-expert MoE layer, while the image transformer remains unchanged.

As shown in Figure 17, this approach continues to accelerate text training loss reduction compared to the dense model and MoT alone (**Figure 17a-b**). Notably, the combined model retains MoT's advantage in the image modality, with comparable training loss and speedup relative to MoT (**Figure 17c-d**). The averaged training loss across both modalities (**Figure 17e-f**) highlights the potential of this approach in handling multi-objective training with a balance between efficiency and performance.

Validation results on representative text-only datasets and image generation tasks confirm the consistency of this approach (**Figure 17g-j**). "MoT + Text MoE-4x" achieves the best text performance, maintaining the efficiency of MoT in the image modality while improving text generation. On the other hand, despite MoE-4x demonstrating lower text training loss than the dense model, it shows little to no improvement[7] in text valtidation losses (**Figure 17g-i, l-n**), further emphasizing the effectiveness of the MoT approach.

Overall, these preliminary results in both the Chameleon and Transfusion settings provide a proof-of-concept for combining the key strengths of MoT and MoE-4x. The integration of MoE-4x into the text modality enhances text performance, while MoT continues to deliver strong results in image generation. Further investigations will explore the scalability and generalizability of this approach across additional tasks and modalities.

---

[7]This observation diverges from our observation in the Chameleon setting (Figure 16), where MoE-4x improved text and image losses during both training and validation. Contrary to conventional understanding of MoE, MoE-4x's training loss gains in Transfusion didn't translate to inference time. The discrepancy may stem from the fact that Transfusion processes discrete text tokens and continuous image tokens, which complicates router generalization during inference. We leave further exploration of integrating MoE and Transfusion to future work.

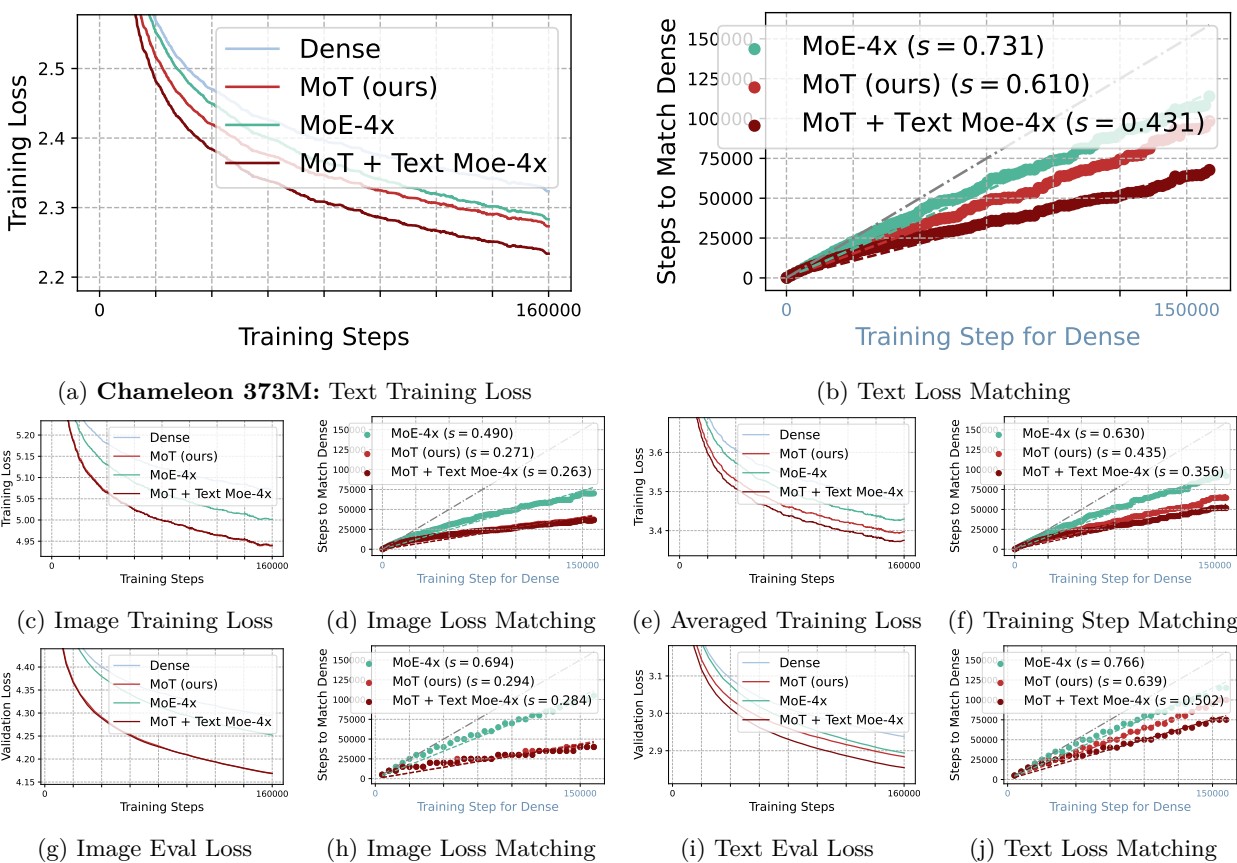

(a) **Chameleon 373M:** Text Training Loss

(b) Text Loss Matching

(c) Image Training Loss

(d) Image Loss Matching

(e) Averaged Training Loss

(f) Training Step Matching

(g) Image Eval Loss

(h) Image Loss Matching

(i) Text Eval Loss

(j) Text Loss Matching

Figure 16: **Combining MoT and MoE-4x in the Chameleon setting.** A hybrid model "MoT + Text MoE-4x" was created by replacing the text feed-forward network in MoT's text transformer tower with MoE-4x. **a-b**, Text training loss reduction is significantly accelerated compared to dense model, MoE-4x, and MoT. **c-d**, Image modality performance benefits of MoT are retained. **e-f**, Averaged training loss across both modalities. **g-j**, Validation losses on Obelisc dataset: "MoT + Text MoE-4x" achieves best text performance while maintaining comparable or slightly improved image performance relative to MoT. Both significantly outperform dense model and MoE-4x.

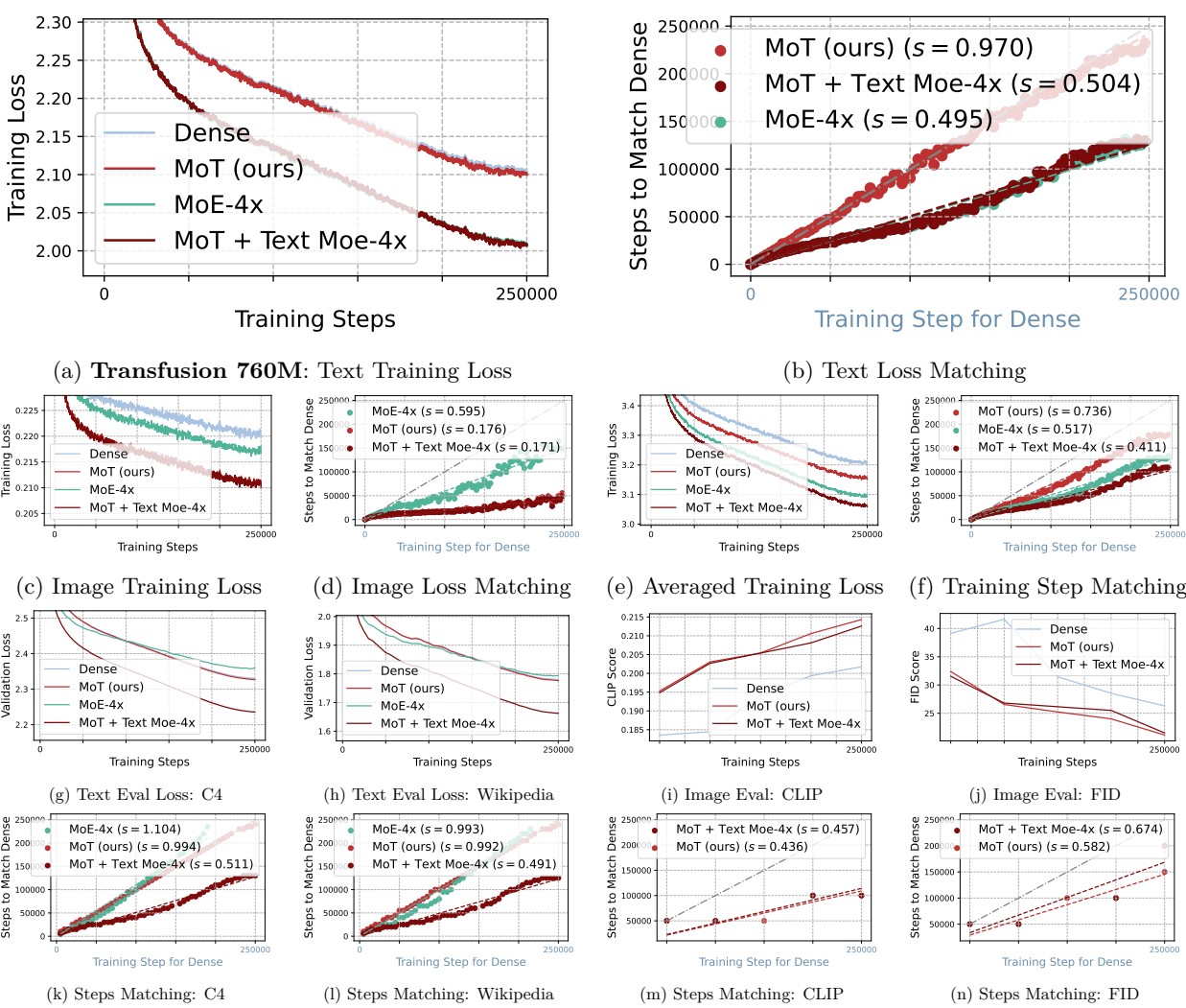

Figure 17: **Combining MoT and MoE-4x in the Transfusion setting. a,b,** Text training loss reduction significantly accelerated compared to dense model and MoT. **c,d,** Image modality performance benefits of MoT retained. **e,f,** Averaged training loss across both modalities. **g-n,** Validation losses on multiple representative text-only datasets and image generation evaluation results: "MoT + Text MoE-4x" achieves best text performance while maintaining comparable or slightly improved image performance relative to MoT. Both significantly outperform dense model and MoE-4x. MoE-4x shows better text training loss than dense model (**a,b**) but minimal improvement in text validation losses (**g-h**, **k-l**).

# 6 ML Systems Aspects of Mixture-of-Transformers

This section highlights a few system properties of MoT, and demonstrates how they translate to real-world benefits in a typical training environment.

## 6.1 Throughput Scaling Properties

**Communication Volume** The modality-based scaling method by MoT maintains a lower Parameter to FLOPs (PpF) ratio. PpF is a crucial metric that governs training throughput in large-scale cloud-based training environments, where distributed learning is highly sensitive to communication overhead (Llama team, 2024; Luo et al., 2020). This is particularly relevant in recent years, as compute capacity has increased significantly faster than network bandwidth (Luo et al., 2018; 2024). Consequently, models with smaller parameter sizes have advantage in terms of training throughput at a large scale.

To quantify this effect, we compare the added parameters to each transformer layer by adding $E$ experts in MoE versus untying $K$ modalities in MoT, in a typical transformer setup with Swiglu FFN layers, with token hidden and embedding dimension both $D$, a feed forward layer has $|FFN| = 3D^2$ parameters, and the KQVO projections in attention layers introduces $|ATTN| = 4D^2$ parameters. We can ignore the normalization layer weights as they are usually in the shape of $D$.

Thus, each MoE layer comprises $E$ feed forward layers and a router paramater of size $|ROUTER| = K \times D$, and a MoE layer has an additional $(E-1) \times |FFN| + |ROUTER| = 3(E-1)D^2 + ED > 3(E-1)D^2$ parameters. In contrast, MoT incurs an added parameter count of $(K-1) \times (|ATTN| + |FFN|) = 7(K-1)D^2$.

Since $E$ can range from a few to even hundreds (Dai et al., 2024; Muennighoff et al., 2024; DeepSeek-AI et al., 2025), a typically much smaller $K$ compared to $E$ implies that MoT can have a lower PpF ratio compared to MoE in general, which can prove beneficial in real world scenarios, as we show later.

**Compute Efficiency** Both MoE and MoT incur additional overheads when routing tokens to the appropriate parameters.

1. MoEs suffer from overheads due to the additional operations of performing Top-K selection, indexing tokens, and scattering and adding expert outputs. These operations are sequentially dependent on each other, making it challenging to hide the resulting latency.

2. In contrast, MoT's overheads primarily come from two sources. First, the CPU-GPU synchronization required for grouping tokens by modality for element-wise projections and reassembling them for attention results in significant overhead, mostly attributed to frequent GPU-CPU synchronization due to masking for specific modalities. Second, the sequential processing of modalities can also lead to underutilization of GPU resources and imbalance, particularly when tokens of different modalities are unevenly distributed across local batches and GPUs.

It's worthwhile to note that the overheads in MoT can be minimized via diligent engineering: for example, caching sequence indices for each modality can substantially reducing indexing costs and needs to be done only once per iteration, as the modalities of tokens do not change. Specialized Group GEMMs (Nvidia) or Megablock-style block sparse matrix multiplication can be employed (Gale et al., 2022) to perform imbalanced projections in one shot across all modalities. Since we did not observe these overheads on the critical path in our training setup, we leave these as future directions to further improve MoT.

## 6.2 Empirical Analysis

We conducted our experiments and system profiling on AWS, using p4de.24xlarge instances equipped with NVIDIA A100 Tensor Core GPUs. Distributed training of all models are powered by Fully Sharded Data Parallel (Zhao et al., 2023) in full shard mode. We enable Pytorch 2 Compiler (Ansel et al., 2024) to optimize the model whenever applicable.

### 6.2.1 Horizontal Scaling—MoT Benefits Increase with GPU Count

We investigated the horizontal scaling capabilities of Mixture-of-Transformers (MoT) in large-scale distributed training. As large language models (LLMs) typically employ larger global batches across increasing numbers of GPUs, we examined MoT scaling trends by varying GPU count during training. Global batch size and total training tokens were increased proportionally to GPU count, while maintaining constant training steps. We conducted the experiments in the Chameleon setting under the 443M model scale.

Figure 18 shows Obelisc dataset evaluation losses as GPU count scales from 16 to 256. MoT performance gains increase substantially with GPU count. For image validation loss, the percentage of training steps required for MoT to match the dense model decreases from 42.1% (16 GPUs) to 21.6% (256 GPUs). For text, this percentage decreases from 75.7% to 50.9%. These results suggest MoT's efficiency and performance benefits grow with increasing pre-training compute resources.

This analysis was conducted under specific AWS infrastructure conditions. Further investigation is needed to generalize these findings across different hardware configurations and training environments.

### 6.2.2 Speed Advantage of MoT in Wall-Clock Time

We investigated the wall-clock time performance of MoT in a specific environment. This analysis is crucial for understanding MoT's practical benefits in real-world training scenarios, where achieving the best model quality within a fixed GPU training time budget is the primary objective. To ensure the accuracy of our claims, we note that our results were obtained using a specific AWS setup, specified above. Therefore, we expect the relative performance of MoE, MoT, and dense models to vary across different clusters. Nevertheless, we believe that our setup represents a typical deployment that leverages cloud computing (e.g., AWS), and thus our experiences and findings will be relevant and beneficial to readers.

Figure 19 illustrates MoT's wall-clock time acceleration over dense Transformer and MoE-4x baselines in terms of GPU training time on 256 GPUs. MoT demonstrates significant improvements in both image and text modalities for a fixed amount of GPU training time. Specifically, MoT matches the dense model's image training loss in 47.2% of the total GPU training time and continues to improve thereafter (Figure 19b). For text, MoT requires 75.6% of the dense model's time to achieve comparable performance (Figure 19d). In contrast, MoE-4x exhibits no speed advantage in the text modality (Figure 19d) and even results in a 1.7x slowdown in the image modality compared to the dense model (Figure 19b). Results are consistent on the evaluation losses on the Obelisc dataset (Figure 19e-h).

### 6.3 Deployment Considerations

So far, our discussion on the MLSys aspect focuses mainly on training. Since mixed-modality requests might not fill a full batch for each modality branch, the inference system could aggregate similar modality operations across multiple requests (when latency budgets allow) to ensure that the specialized GEMM kernels are utilized efficiently, i.e, via dynamic batching (git; Yu et al., 2022) or continuous batching (any). Orthogonally, we note that when sequence length is not prohibitively long, or when the input modalities are mostly uniform, the modality with fewer number of tokens can be padded to align with the modality with the most tokens, thereby we can compute all modalities' GEMM at once with a single batched matrix multiplication, trading off slightly wasted compute with lower latency.

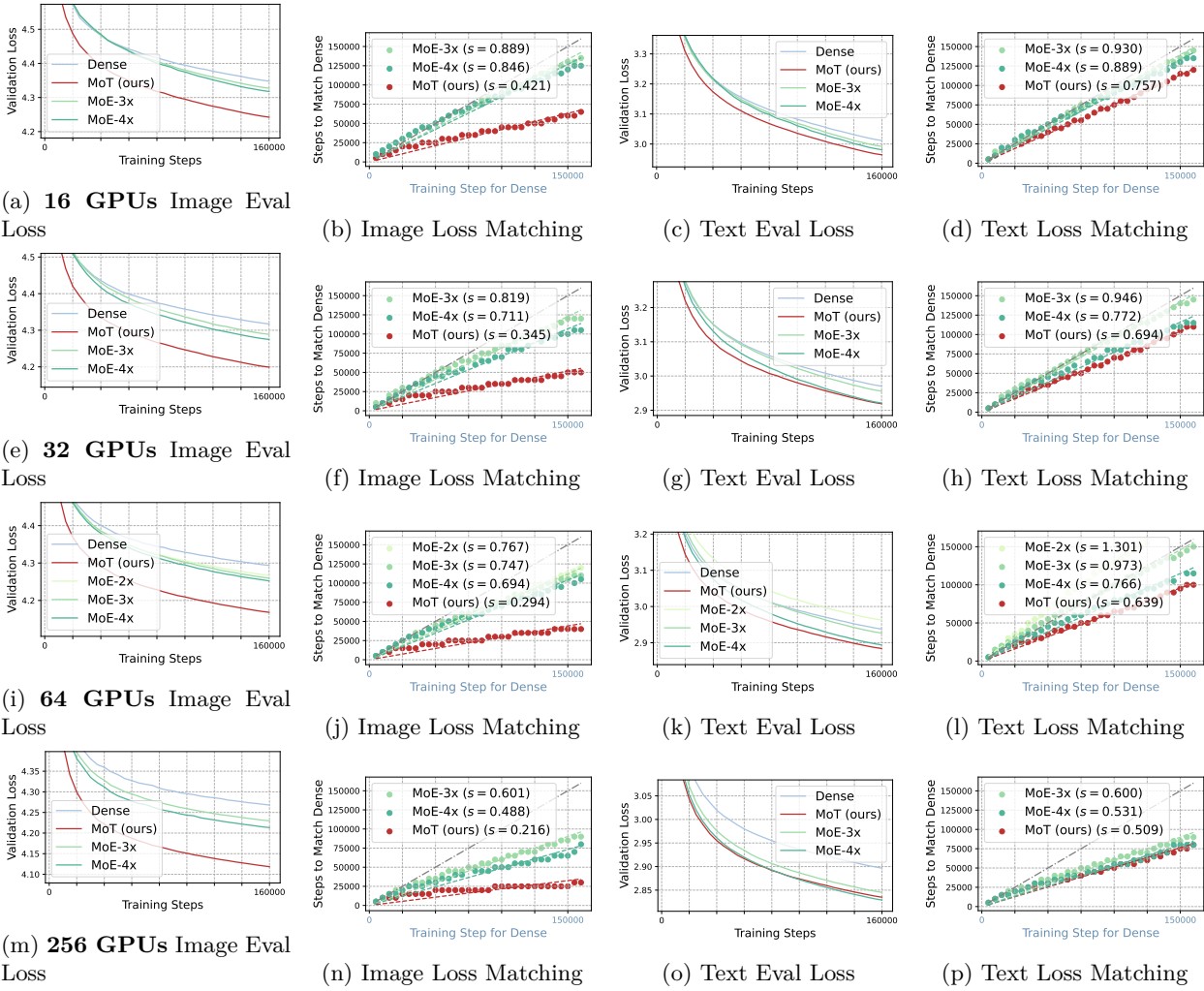

Figure 18: **Horizontal scaling — The benefits of MoT increase with the number of GPUs (Chameleon setting, 443M model scale).** As modern LLMs are typically trained on larger global batches using more GPUs, we conduct a pilot study on the scaling trends of MoT by varying the number of training GPUs. This effectively scales the global batch size and the total number of training tokens, while keeping the number of steps constant. Shown are the evaluation losses on the Obelisc dataset. The performance gains of MoT increase substantially as the number of GPUs grows. For instance, when scaling from 16 to 256 GPUs, the percentage of steps required for MoT to match the image validation loss of the dense model (trained with the same number of GPUs) decreases from 42.1% to 21.6%, and for text validation loss, from 75.7% to 50.9%. This suggests that scaling pre-training compute resources further enhances the efficiency and performance gains of MoT.

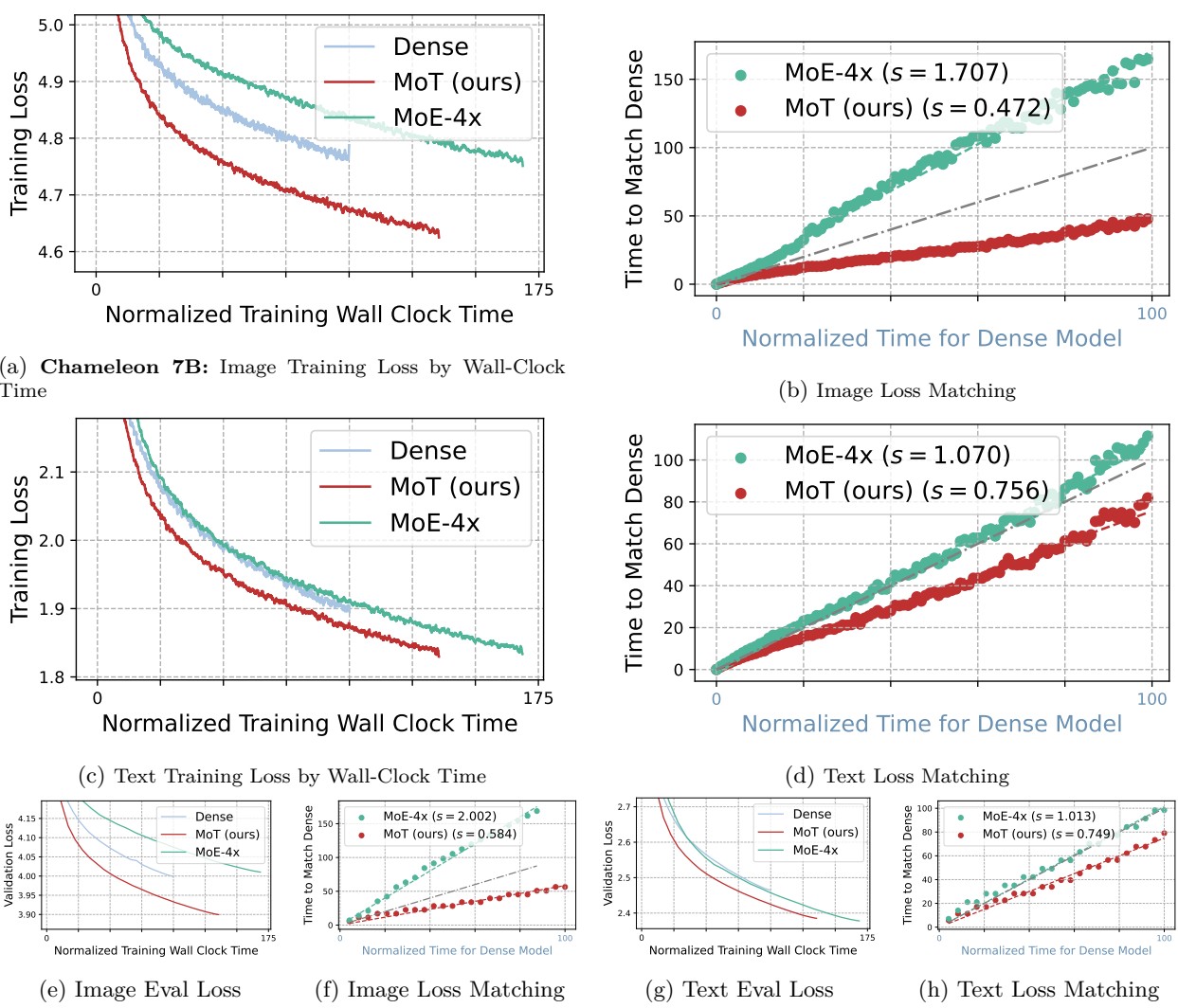

(a) **Chameleon 7B:** Image Training Loss by Wall-Clock Time

(b) Image Loss Matching

(c) Text Training Loss by Wall-Clock Time

(d) Text Loss Matching

(e) Image Eval Loss    (f) Image Loss Matching    (g) Text Eval Loss    (h) Text Loss Matching

Figure 19: **Speed advantage of MoT in wall-clock time (Chameleon Setting).** For a fixed amount of GPU training time, MoT significantly outperforms both the dense Transformer baseline and MoE-4x. MoT matches the image training loss of the dense model in just 47.2% of the GPU training time, with continued improvement. For the text modality, MoT requires 75.6% of the time to achieve the same quality. In contrast, MoE-4x shows no speed advantage in the text modality and results in a 1.7x slowdown in the image modality. Evaluation losses on the Obelisc dataset show consistent results.

# 7 Related Work

## 7.1 Foundation Models for Multi-Modal Generation

Recent advances in large language models (LLMs) have extended to multi-modal applications. Early multi-modal LLMs focused on understanding rather than generation, using late fusion techniques to merge separately encoded images and text (Alayrac et al., 2022; Liu et al., 2023; Laurençon et al., 2023; Chen et al., 2022). While benefiting from lightweight training, these models lacked multi-modal generation capabilities.

To enable multi-modal generation, a key strategy involves tokenizing non-text modalities into discrete sequences (Aghajanyan et al., 2022; Yu et al., 2023; Bao et al., 2021; Ramesh et al., 2021; Liu et al., 2024c) (Figure 2a). For instance, Chameleon (Chameleon Team, 2024) and related approaches (Aghajanyan et al., 2022) tokenize images into 1,024 discrete tokens using pretrained models like VQGAN (Esser et al., 2021), training over combined text-image token sequences. Similar tokenization has been applied to speech (Nguyen et al., 2024). Recent models like Transfusion (Zhou et al., 2024) have explored continuous image tokens and diffusion-based loss functions to enhance visual generation quality. Our proposed mixture of transformers method is compatible with these approaches and can be integrated as a drop-in replacement for dense transformer architectures. We demonstrate substantial improvements across diverse multi-modal settings, including both Chameleon (Chameleon Team, 2024) and Transfusion (Zhou et al., 2024).

## 7.2 Sparse Architectures for Multi-Modal Generation

Sparse architectures, particularly Mixture of Experts (MoE), have shown promise in text-based models, allowing dynamic parameter selection for each input (Jacobs et al., 1991; Eigen et al., 2013; Shazeer et al., 2017; Lepikhin et al., 2020; Fedus et al., 2022; Jiang et al., 2024; Sukhbaatar et al., 2024). Recent efforts have adapted MoE for multi-modal tasks, addressing the challenges posed by inherent feature space gaps between modalities (Wang et al., 2022; Shen et al., 2023; Bao et al., 2022a; Long et al., 2023; Lin et al., 2024). These approaches suggest that modality-specific parameter allocation can improve performance by addressing distinct data type (i.e., modality) characteristics (Liang et al., 2022). Different from previous works, in this paper, we propose the Mixture-of-Transformers (MoT) framework, which generalizes the MoE concept by decoupling all non-embedding parameters within the transformer architecture. MoT consistently outperformed MoE in multi-modal pretraining when the amount of total parameters is controlled (Figure 16) and demonstrated complementarity with MoE-4x (Figure 17).

While recent works have extended MoE beyond feedforward layers to attention mechanisms (Wang et al., 2023; Shen et al., 2024; Liu et al., 2024b), our approach differs in several key aspects. Unlike CogVLM (Wang et al., 2023), which is limited to generating text outputs, MoT is capable of both image and text generation. Concurrent to our work, Playground v3 (PGv3) (Liu et al., 2024b) integrates a DiT-style image transformer with Llama3-8B as the text backbone using global self-attention, and achieves state-of-the-art performance in text and image generation. During training, the text LLM is frozen and only the image transformer component is updated. While both CogVLM and PGv3 conduct multi-modal training on top of a pre-trained LLM, we establish MoT as a general sparse architecture that can be trained from scratch. MoT also decouples every non-embedding parameter across transformer layers, including layer normalization, whereas previous approaches maintain shared layernorm parameters. Our findings position MoT as a flexible and scalable solution for multi-modal pretraining, demonstrating its potential to complement MoE-based architectures and offering a pathway for more computationally efficient large-scale multi-modal models.

# 8 Conclusion

In this work, we present Mixture-of-Transformers (MoT), a sparse and scalable architecture designed to address the computational challenges of multi-modal model pretraining. By decoupling non-embedding parameters by modality and retaining global self-attention across multi-modal sequences, MoT optimizes modality-specific processing while preserving cross-modal interactions. Our experiments demonstrate that MoT achieves significant reductions in training costs across various settings and model scales. In the Chameleon and Chameleon+Speech settings, MoT matched or exceeded the performance of dense base-

lines while using substantially fewer FLOPs. Furthermore, MoT maintained these improvements in a more complex setting (Transfusion), where distinct training objectives were applied to different modalities, demonstrating consistent efficiency gains and enhanced performance in tasks such as image generation. In addition to FLOP reductions, system profiling highlights the practical benefits of MoT, including reductions in wall-clock time for both text and image tasks. When scaled across GPUs, MoT demonstrated further improvements, indicating its suitability for large-scale distributed training environments. Preliminary results combining MoT with Mixture-of-Experts (MoE-4x) suggest the potential for hybrid models that further improve performance without increasing computational costs. These findings suggest that MoT could serve as an effective framework for future multi-modal LLMs, enabling more efficient large-scale training while maintaining competitive performance across diverse modalities.

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

# A  Tranfusion: Preliminaries

## A.1  Diffusion for Image Generation

Diffusion models have emerged as a powerful class of generative models capable of producing high-fidelity data across various modalities. These models utilize a Markov chain that progressively adds Gaussian noise to data in a forward process and then learns to reverse this process to generate new data samples.

In the **forward diffusion process**, the data $\mathbf{x}_0$ is perturbed over $T$ timesteps by sequentially adding Gaussian noise. The transition from $\mathbf{x}_{t-1}$ to $\mathbf{x}_t$ is defined by the conditional probability distribution: $q(\mathbf{x}_t \mid \mathbf{x}_{t-1}) = \mathcal{N}(\mathbf{x}_t; \sqrt{\alpha_t}\,\mathbf{x}_{t-1}, (1 - \alpha_t)\mathbf{I})$, where $\alpha_t \in (0,1)$ controls the rate of noise addition at each timestep $t$. The cumulative product of $\alpha_t$ up to timestep $t$ is denoted by $\bar{\alpha}_t = \prod_{s=1}^{t} \alpha_s$. Using this notation, we can express $\mathbf{x}_t$ directly in terms of the original data $\mathbf{x}_0$: $\mathbf{x}_t = \sqrt{\bar{\alpha}_t}\,\mathbf{x}_0 + \sqrt{1 - \bar{\alpha}_t}\,\boldsymbol{\epsilon}$, where $\boldsymbol{\epsilon} \sim \mathcal{N}(\mathbf{0}, \mathbf{I})$ is standard Gaussian noise. As $t$ approaches $T$, the data distribution transitions towards an isotropic Gaussian distribution.

In the **reverse diffusion process**, the goal is to recover the original data $\mathbf{x}_0$ from the noisy observation $\mathbf{x}_T$ by iteratively denoising. The reverse process is parameterized by a neural network $\boldsymbol{\epsilon}_\theta(\mathbf{x}_t, t, c)$ trained to predict the added noise at each timestep, where c is extra context, such as text prompt. The denoising step can be expressed as: $\mathbf{x_{t-1}} = \frac{1}{\sqrt{\alpha_t}}(\mathbf{x_t} - \frac{1 - \alpha_t}{\sqrt{1 - \bar{\alpha}_t}}\,\boldsymbol{\epsilon}_\theta(\mathbf{x}_t, t, c)) + \sigma_t\,\boldsymbol{z}$, where,$\sigma_t$ is the standard deviation of the noise added during the reverse step, and $\boldsymbol{z} \sim \mathcal{N}(\mathbf{0}, \mathbf{I})$ is auxiliary noise introduced for stochasticity in the sampling. The neural network $\boldsymbol{\epsilon}_\theta$ is trained by minimizing the objective function $L_{\text{DDPM}} = \mathbb{E}_{\mathbf{x}_0, \boldsymbol{\epsilon}, t}\left[\|\boldsymbol{\epsilon} - \boldsymbol{\epsilon}_\theta(\mathbf{x}_t, t, c)\|^2\right]$. Once optimized, a new data point $\mathbf{x}_0$ can be sampled by initializing $\mathbf{x}_T \sim \mathcal{N}(\mathbf{0})$, following the above denoising steps.

In this work, we use cosine scheduler (Nichol & Dhariwal, 2021) to set the value of $\alpha_t$. We show qualitative image generation results from the 7B model. In this case, we use Classifier-free guidance (CFG) (Ho & Salimans, 2022) to improve generation by contrasting the prediction of the model conditioned on the context $c$ with the unconditioned prediction.

To reduce computational requirements, we adopt latent diffusion models (LDMs) (Rombach et al., 2022) , which perform the diffusion process in a lower-dimensional latent space (e.g. represent every $8{\times}8$ pixel patch as an 8-dimensional vector.) rather than directly in the high-dimensional data space. Specifically, we first encode the original data $\mathbf{x}_0$ into a latent representation $\mathbf{z}_0$ using a Variational autoencoders (VAEs) (Kingma & Welling, 2013)). The diffusion (forward and reverse) process is then applied to $\mathbf{z}_0$, significantly reducing computational cost due to the lower dimensionality of the latent space. This approach allows efficient training and sampling while preserving the quality and fidelity of the generated multimodal outputs.

## A.2  Transfusion Model Architecture

The model primarily consists of a single transformer to processes the combined sequence regardless of modality. We follow Llama's architecture (Touvron et al., 2023a) to build transformer layers, and add lightweight modality-specific module to map the inputs into a shared high-dimensional vector space $\mathbb{R}^d$. For text, embedding matrices convert input integers to vectors and output vectors back into token probabilities. For images, we employ a U-Net to compress local windows of $2 \times 2$ patch vectors in VAE latent space into single vectors suitable for the transformer (and vice versa). In this setting, an image is represented as 256 continous tokens. The transformer uses a hybrid attention mechanism: causal attention is applied across the entire sequence to preserve the autoregressive property, while bidirectional attention is used within each image to capture intra-image dependencies. This means that image patches can attend to all other patches within the same image but only to preceding tokens or image patches outside their own image.

The model is trained by minimizing a combined loss function:

$$\mathcal{L}_{\text{Transfusion}} = \mathcal{L}_{\text{LM}} + \lambda \cdot \mathcal{L}_{\text{DDPM}}, \tag{4}$$

where $\lambda$ is a balancing coefficient. The language modeling loss $\mathcal{L}_{\text{LM}}$ is computed per token, encouraging the model to predict the next token in the sequence. The diffusion loss $\mathcal{L}_{\text{DDPM}}$ is computed per image. We set

the $\lambda$ coefficient in the Transfusion objective to 5 following preliminary experiments; we leave further tuning of $\lambda$ to future work.

During inference, the model alternates between language modeling and diffusion sampling modes. In language modeling mode, it generates text by sequentially sampling tokens from the predicted probability distribution. When a *beginning of image* (BOI) token is generated, the model switches to diffusion mode. In this mode, pure noise $\mathbf{x}_T$ is appended to the input sequence as a series of image patches corresponding to the desired image size. The model then iteratively denoises this input. Once the diffusion process concludes, an *end of image* (EOI) token is appended to the sequence, and the model returns to language modeling mode. At the mean time, an image is generated using a VAE decoder. This seamless switching mechanism allows Transfusion to generate sequences containing any mixture of text and images, leveraging shared parameters and modality-specific processing within a unified architecture.

# B MoT Transfusion Fine-tuning Results

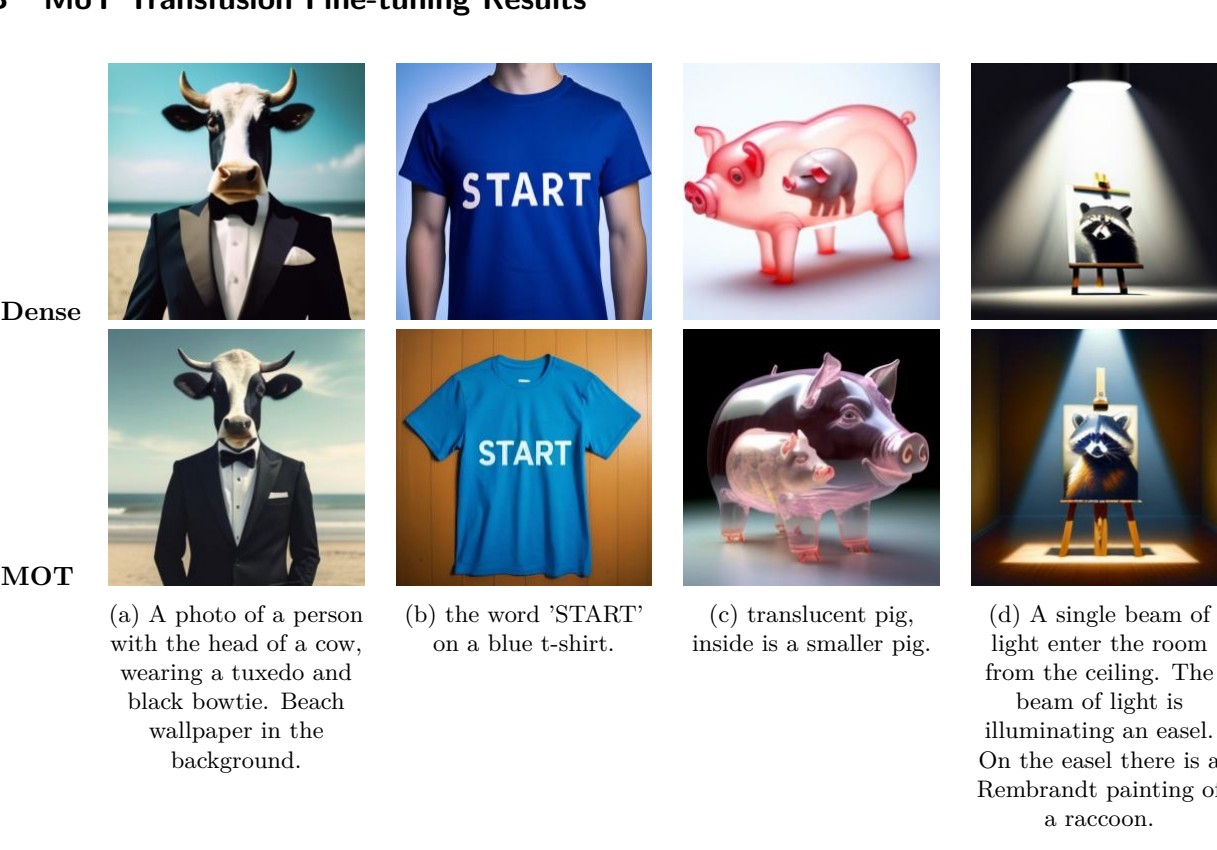

(a) A photo of a person with the head of a cow, wearing a tuxedo and black bowtie. Beach wallpaper in the background.

(b) the word 'START' on a blue t-shirt.

(c) translucent pig, inside is a smaller pig.

(d) A single beam of light enter the room from the ceiling. The beam of light is illuminating an easel. On the easel there is a Rembrandt painting of a raccoon.

Figure 20: Example easy prompts

We compare the image generation capabilities of fine-tuned TransFusion MoT and dense models by prompting them with a variety of text inputs, as illustrated in Figures 21, 20, and 22. In Figure 21, both MoT and dense fine-tuned models successfully follow the prompts. However, in Figure 20, the MoT fine-tuned model demonstrates superior performance, producing images that are either more visually appealing or more faithful to the prompts. In Figure 22, both models struggle to perfectly follow the text prompts and fail to capture all the details accurately. Our study suggests that text faithfulness can greatly improve with extended training and we leave it future work to scale up training with bigger model and more data.

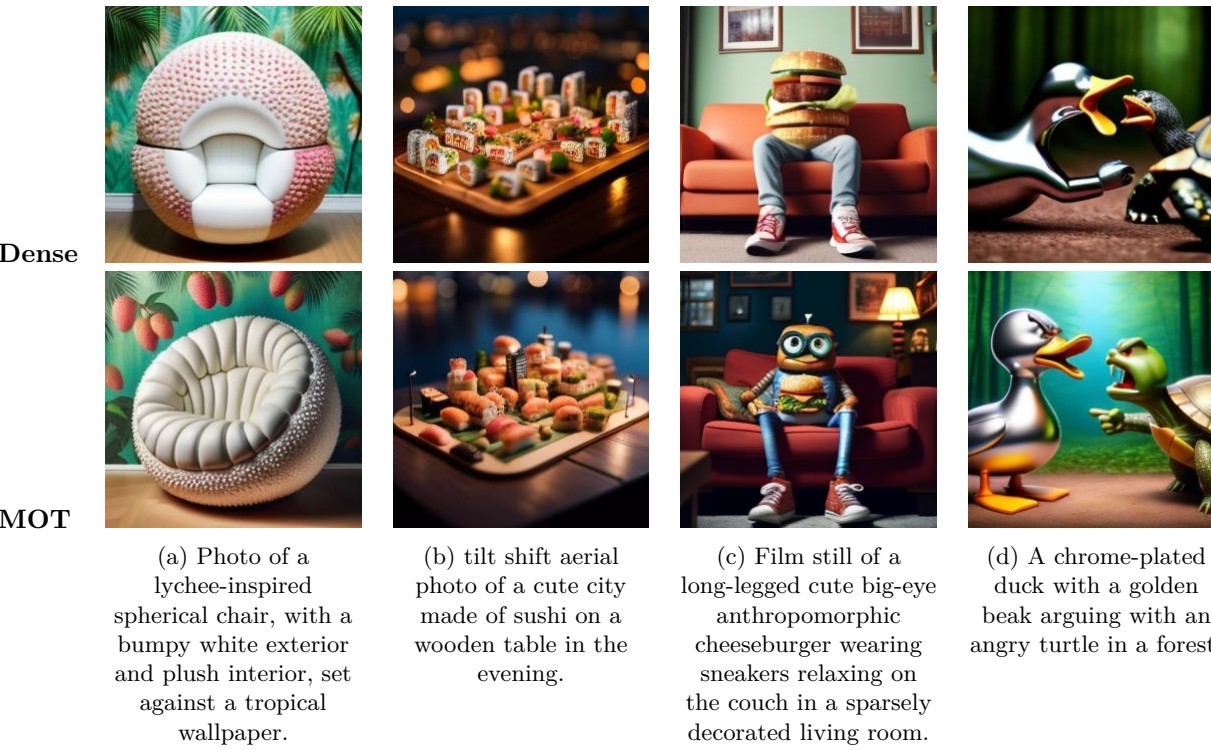

**Dense**

**MOT**

(a) Photo of a lychee-inspired spherical chair, with a bumpy white exterior and plush interior, set against a tropical wallpaper.

(b) tilt shift aerial photo of a cute city made of sushi on a wooden table in the evening.

(c) Film still of a long-legged cute big-eye anthropomorphic cheeseburger wearing sneakers relaxing on the couch in a sparsely decorated living room.

(d) A chrome-plated duck with a golden beak arguing with an angry turtle in a forest.

Figure 21: Example prompts where MOT yields better image generation than Dense

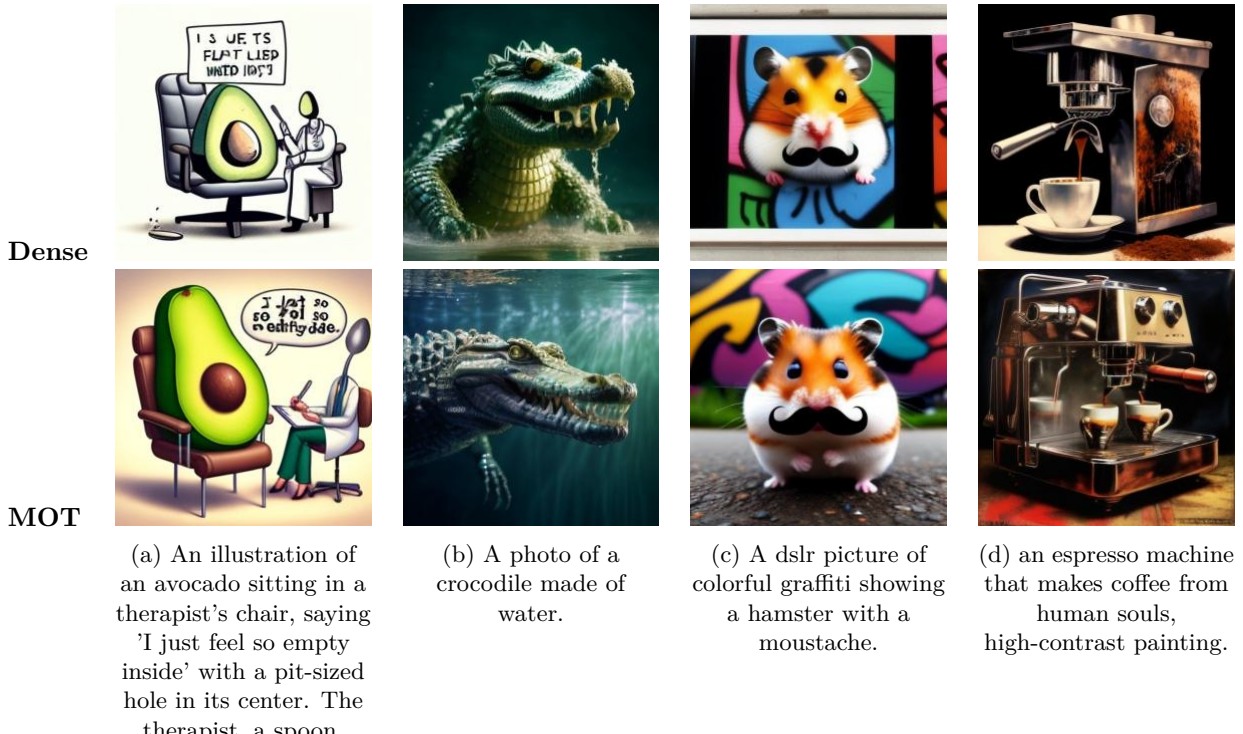

**Dense**

**MOT**

(a) An illustration of an avocado sitting in a therapist's chair, saying 'I just feel so empty inside' with a pit-sized hole in its center. The therapist, a spoon, scribbles notes.

(b) A photo of a crocodile made of water.

(c) A dslr picture of colorful graffiti showing a hamster with a moustache.

(d) an espresso machine that makes coffee from human souls, high-contrast painting.

Figure 22: Example hard prompts

## C    Supplementary Figures

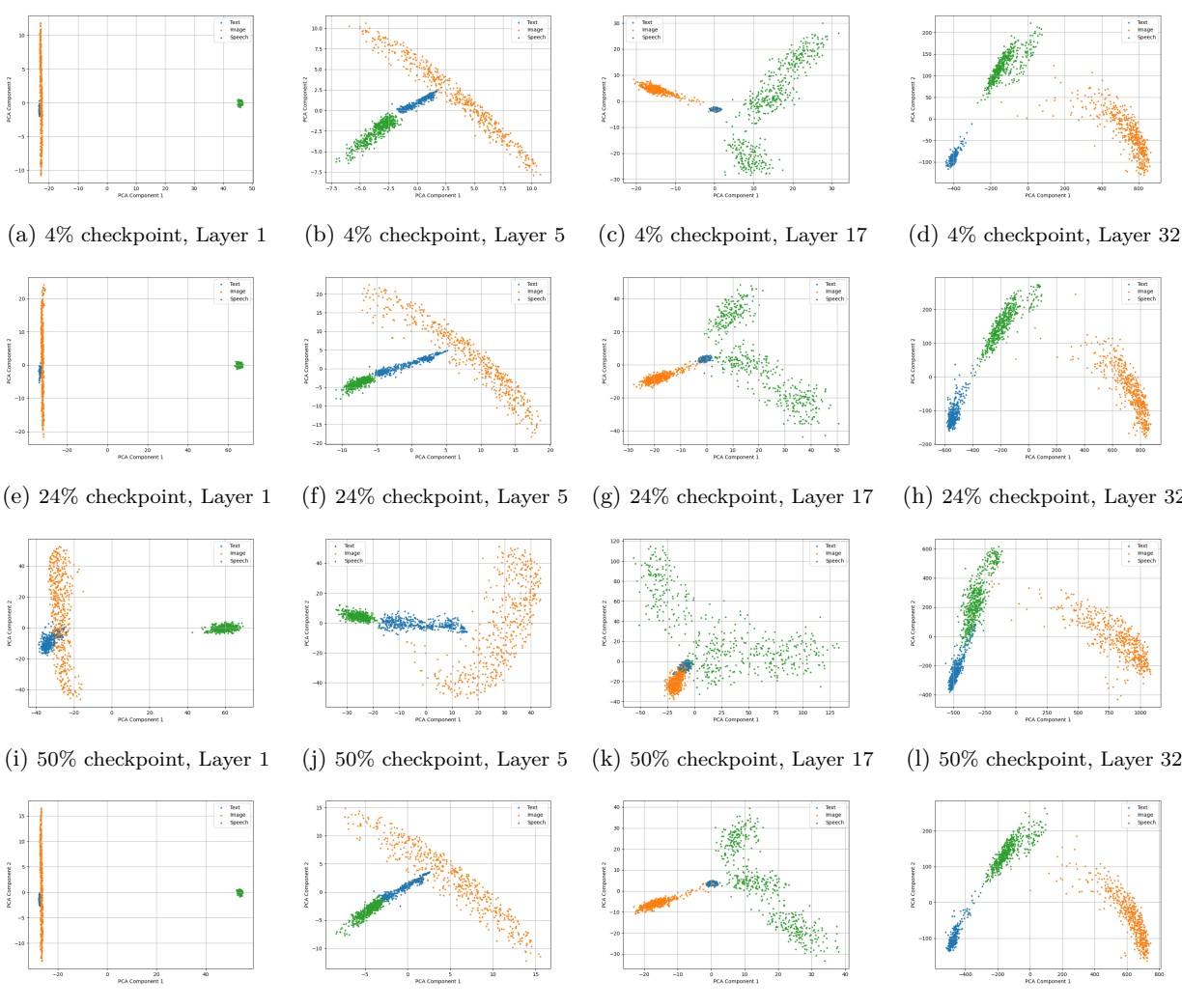

(a) 4% checkpoint, Layer 1    (b) 4% checkpoint, Layer 5    (c) 4% checkpoint, Layer 17    (d) 4% checkpoint, Layer 32

(e) 24% checkpoint, Layer 1    (f) 24% checkpoint, Layer 5    (g) 24% checkpoint, Layer 17    (h) 24% checkpoint, Layer 32

(i) 50% checkpoint, Layer 1    (j) 50% checkpoint, Layer 5    (k) 50% checkpoint, Layer 17    (l) 50% checkpoint, Layer 32

(m) 100% checkpoint, Layer 1  (n) 100% checkpoint, Layer 5  (o) 100% checkpoint, Layer 17    (p) 100% checkpoint, Layer 32

Figure 23: **Visualization of latent feature space for Chameleon+Speech 7B Dense model across training checkpoints and layers.** Principal Component Analysis (PCA) of model activations shows clustering by modality (text, speech, image) at different stages of training (4%, 24%, 50%, 100% checkpoints) and across layers (Layer 1, Layer 5, Layer 17, Layer 32). The PCA plots show that different modalities consistently occupy distinct regions of the feature space. This natural clustering highlights the inherent differences between modalities, suggesting that they are processed differently by the model. These findings motivate the need for decoupled weights in our Mixture-of-Transformers architecture, where modality-specific parameters can better capture and leverage the distinct statistical properties of each modality, leading to improved performance and efficiency compared to a dense baseline.

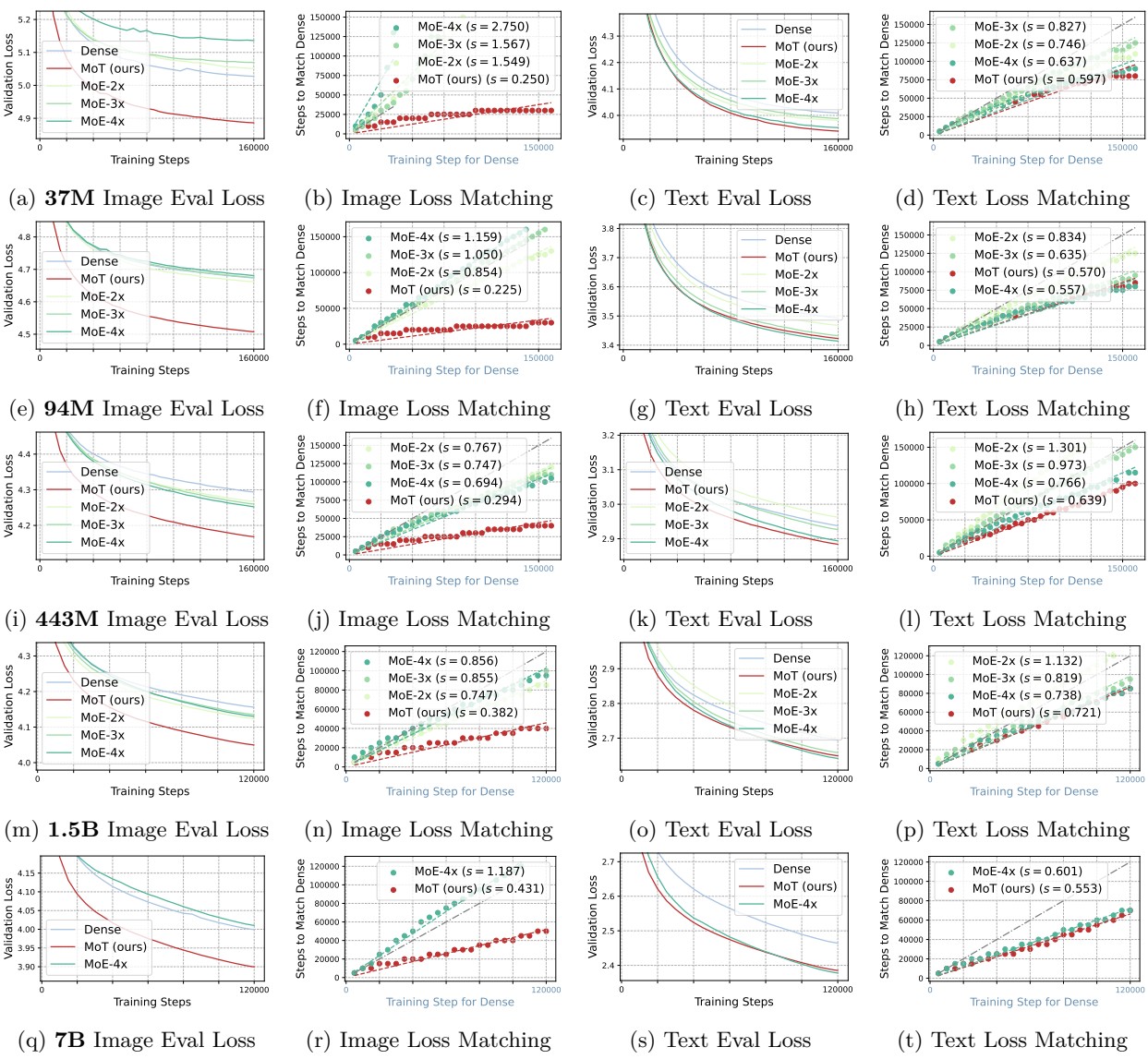

Figure 24: **Training and validation losses for image and text modalities across model scales (37M, 94M, 443M, 1.5B, 7B) in the Chameleon setting evaluated on the Obelisc dataset.** For the image modality, MoT consistently delivers a substantial speedup relative to the dense model and MoE-4x, with the advantage growing across scales. In contrast, MoE-4x exhibits diminishing gains as the model scales increase, particularly at 7B, where the benefits disappear in the image modality. In the text modality, both MoT and MoE-4x outperform the dense model, with MoT demonstrating comparable or slightly better performance. FLOPs-controlled across all runs in the same model scale and pre-trained from scratch.

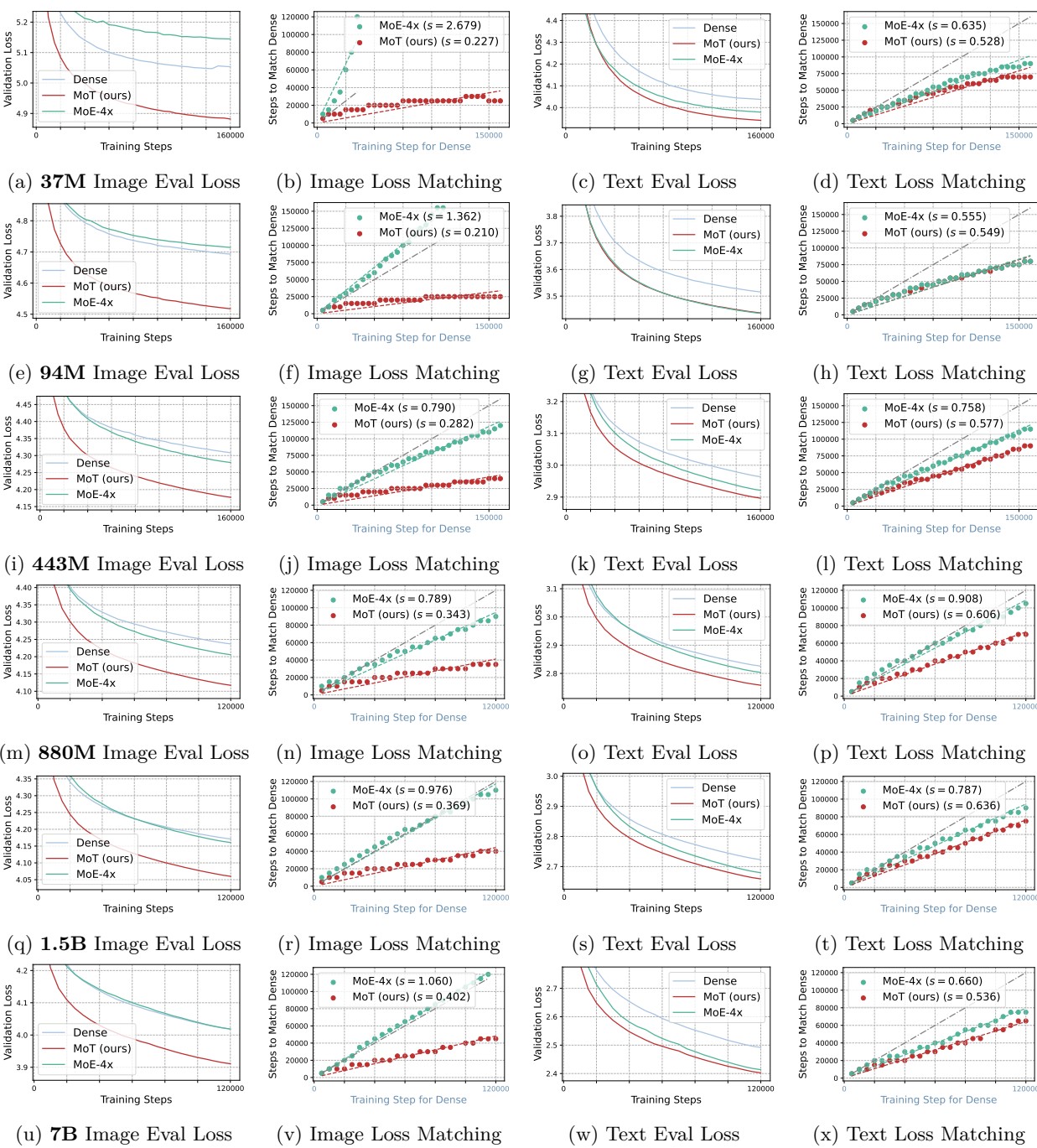

Figure 25: **Training and validation losses for image and text modalities across model scales (37M, 94M, 443M, 1.5B, 7B) in the Chameleon+Speech setting evaluated on the Obelisc dataset.** MoT exhibits consistent and significant improvements in validation loss for the image and text modalities, demonstrating its efficiency and robustness across scales. FLOPs-controlled across all runs in the same model scale and pre-trained from scratch.

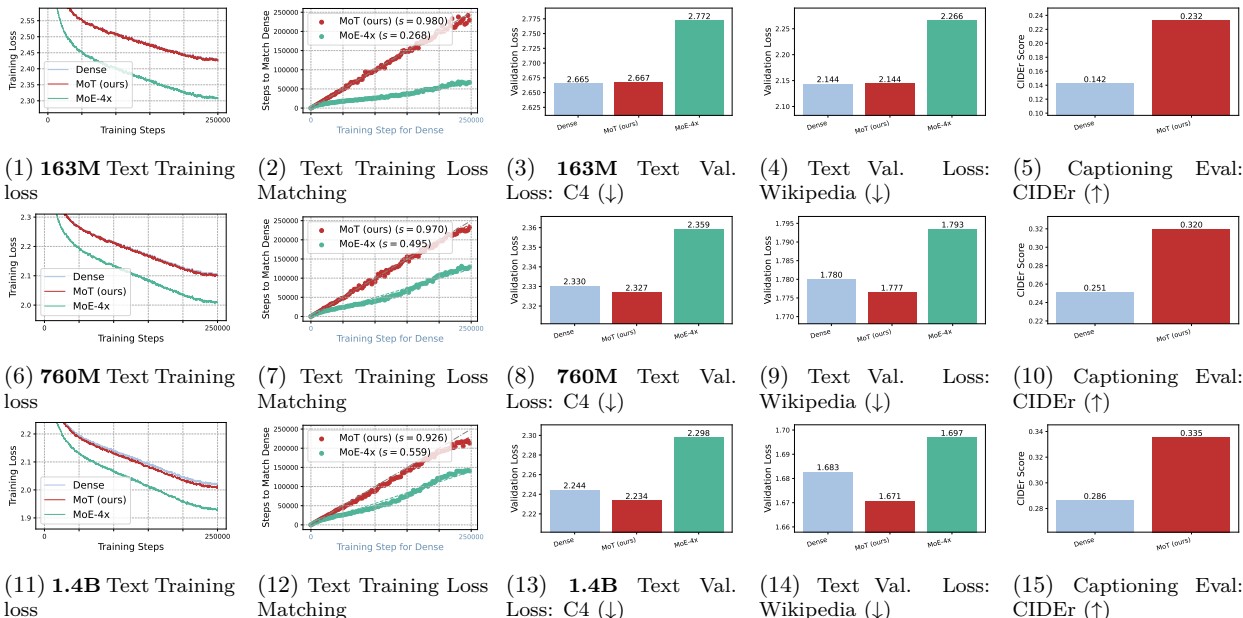

Figure 26: **Text Modality-specific training loss and step matching plots in Transfusion setting across model scales.** For text modality, MoT matches dense model in training and validation loss on C4 and Wikipedia datasets, with improved generalization in captioning tasks (CIDEr score). MoE-4x shows unstable performance: lower training losses but poorer generalization than dense model on text evaluation metrics. Model sizes for sparse models indicate activated parameters. All experiments FLOPs-controlled and pre-trained from scratch.

