# OpenReview forum: "Mixture-of-Transformers: A Sparse and Scalable Architecture for Multi-Modal Foundation Models"
_TMLR — Accepted by TMLR_

### Review · Reviewer_KhiW · 2025-01-10

**Summary Of Contributions:**

This paper explores a new model architecture, Mixture of Transformers for multimodal foundation models. The core design is using independent trainable transformer for the tokens from different modalities, but conduct cross-modality computation with attention activations.

This proposed model achieved very promising results on all three modalities, i.e. text, speech, and vision. For the vision part, authors consider both discrete token modeling (Chamelon) and also diffusion modeling (Transfusion). And achieved promising results on both of them.

The comprehensive ablation studies further supports the effectiveness of the proposed model.

**Audience:**

Yes

**Claims And Evidence:**

Yes

**Requested Changes:**

See weakness

**Strengths And Weaknesses:**

Strength:
1) As mentioned in the summary part, the experiments are pretty comprehensive and the results are very strong. The LOO ablation study in Section 4 is also very helpful to understand the contribution of this model arch.
2) Balancing different modalities is challenging in general, including the data resource constrains, training stabilities.... This idea has a great potential to alleviate this problem.
3) Although this idea introduce more parameters, the final model is actually not fully required for many of the downstream tasks. For instance, for the text only applications, people can deploy only a subset of the model (text-only parameters) so that greatly reduce the memory requirement. However, MoE-based model must deploy the whole model for inference. (Seems that authors did not mention this in the draft, please add this into your paper if agree.)
4) It is good to see the improvement from MoE and MoT can be stacked together.



Weakness:
1) (Minor) Could authors please fix the white background of the font in their Figure 1? It is a bit weird.
2) As stated in the paper, there is a relatively large overhead for gathering the tokens for each modality. And this paper did not show how large the overhead is in terms of the training throughput (Tokens/Second) or MFU. I understand that, due to faster convergence, it is still better, but it is very important to show how large the overhead is and maybe people can get some ideas to speed it up.
3) Seems that the MoE baseline is a bit weak? The EC is good but I dont think a well-tuned MoE should be worse than Dense given the same compute budget. Any further justification on this?

---

### Review · Reviewer_fTvF · 2025-01-16

**Summary Of Contributions:**

The Mixture-of-Transformers (MoT) architecture builds multi-modal foundation models for text, images, and speech. MoT uses modality-aware sparsity, decoupling parameters like feed-forward networks and attention projection matrices, while applying global self-attention. Experiments show MoT achieves comparable performance to dense models with reduced computational costs, in settings including: Chameleon (text and images), Chameleon+Speech, and Transfusion (autoregressive text and diffusion images). Ablation studies and analyses were also conducted.

**Audience:**

Yes

**Broader Impact Concerns:**

• Reduced Computational Costs: The primary positive impact of MoT is the potential to reduce the computational cost of training large multi-modal models. This can significantly democratize access to advanced AI, making it accessible to a wider range of researchers and developers.

• Energy/Power Consumption: By reducing computational costs, the MoT architecture may also lead to a reduction in the energy consumption associated with training large models, contributing to more sustainable AI development.

• Potential Misuse: As with any advanced AI technology, there is a potential for misuse. The ability to generate convincing text, images, and speech could be exploited for malicious purposes, such as generating misinformation or creating deepfakes. It is important to consider the ethical implications and develop safeguards to prevent misuse.

• Accessibility: By making multi-modal models more efficient, MoT could enable the deployment of advanced AI applications on devices with limited resources, increasing their accessibility to a broader audience.

• Data Bias: The models are still trained on existing datasets, which may contain biases. While the architecture itself may be bias-neutral, it may still be susceptible to perpetuating biases in its training data. Further research is needed to mitigate this.

**Claims And Evidence:**

Yes

**Requested Changes:**

• More detailed analysis of MoE: Provide further analysis into the instability of the MoE models and the conditions under which MoE performs poorly compared to both dense and MoT models. This analysis may shed light on the mechanisms by which MoT is able to perform so well and why similar gains are not achieved in MoE models.

• Parameter Count Comparisons: Explicitly compare the parameter counts between the dense, MoE, and MoT models to further clarify the efficiency gains in MoT. It is unclear from the results whether the total number of parameters are similar, even though the FLOPs is controlled in the experiment.

• Text performance improvement: Investigate further how MoT could be improved for text-based tasks, particularly in the Transfusion setting. Since MoT performs so well in image generation, more explicit analysis into what contributes to the lack of improvement on text tasks would be helpful.

• Generalization Analysis: Explore the generalization capabilities of MoT in settings where the distribution of data during evaluation differs substantially from the training data. Test MoT on different datasets to further assess how well it can generalize in different domains and settings.

• Real-World Validation: Conduct further real-world validation to understand how the MoT model performs outside of the controlled lab settings. Such experiments would help make the claims of the paper even more robust.

• Hyper parameter Tuning: Provide more details on hyper parameter tuning for the MoT and baseline models, especially for the diffusion objective, and justify the choice of hyper parameters used in the study.

**Strengths And Weaknesses:**

• Strengths:

◦Novel Architecture: The MoT architecture is a novel and significant contribution to the field of multi-modal models, presenting a distinct approach from prior methods, and is shown to outperform MoE models while using fewer FLOPs.

◦ Modality-Aware Sparsity: The core concept of modality-aware sparsity is well-motivated by the inherent differences in how modalities are processed, and is supported by PCA analysis of latent feature spaces, which show distinct clustering by modality.

◦ Scalability and Efficiency: The MoT architecture demonstrates significant efficiency gains by achieving comparable or better performance than dense models while using fewer FLOPs. The paper also shows that the benefits of MoT increase with the number of GPUs during training.

◦ Strong Experimental Results: The paper provides strong empirical results across a variety of model sizes and multi-modal tasks, including text, image, and speech generation, showing that MoT is versatile for large-scale multi-modal generative tasks.

◦ Comprehensive Analysis: The inclusion of ablation studies, a leave-one-out analysis, and exploration of combining MoT with MoE provide a thorough understanding of the MoT architecture and its potential.

◦ Clear Presentation: The paper presents complex information in a clear and well-structured manner. Figures and tables are effective for communicating results and key concepts.

•Weaknesses:

◦ Limited Generalization: While MoT shows strong performance gains, the paper notes that MoE-4x's performance can be unstable when the evaluation data distribution differs significantly from the training data, which may lead to uneven token distribution among experts. This highlights the potential for MoT to also struggle with more significant data distribution changes.

◦ Inconsistent MoE Performance: The paper mentions that MoE-4x shows mixed performance across all scales, sometimes underperforming the dense baseline in speech validation loss despite outperforming it in pre-training loss. This suggests a need for further investigation into why MoE performance is unstable.

◦ Text Performance: The text performance of MoT in the Transfusion setting shows little improvement across scales. This is a potential area for future research.

◦ Parameter count: Although the model shows great improvements in training with fewer FLOPs, parameter counts are not explicitly compared with other models in the study.

◦ Limited Real-World Validation: The experiments are primarily conducted in a controlled environment. The paper would benefit from further real-world validation and testing in more diverse deployment environments.

---

### Review · Reviewer_oCiV · 2025-02-07

**Summary Of Contributions:**

This paper introduces Mixture-of-Transformers (MoT), a sparse architecture for multi-modal foundation models. The key idea is to decouple non-embedding parameters by modality while applying global self-attention across all tokens in a unified sequence. This design addresses the high FLOPs demands of multi-modal training by selectively activating modality-specific parameters (feed-forward layers, attention projections, and layer norms), thereby reducing computation.

The work thoroughly evaluates MoT in three settings:

1. Chameleon
2. Chameleon+Speech
3. Transfusion

Experiments at multiple model scales consistently show that MoT matches or exceeds the performance of dense baselines while requiring substantially fewer FLOPs. The paper also provides system profiling to demonstrate training speedups in wall-clock time. Additionally, there are discussions on combining MoT with Mixture-of-Experts (MoE) and ablations on modality separation in different parts of the transformer (FFNs, attention matrices, layer norms).

Overall, the paper contributes:
- A new multi-modal sparse design with global self-attention.
- Comprehensive experimental evaluation.
- System-oriented analysis for communication overheads and practical throughput benefits.

**Audience:**

Yes

**Broader Impact Concerns:**

No immediate ethical red flags beyond standard concerns for generative models are evident.

**Claims And Evidence:**

Yes

**Requested Changes:**

1. **Extended Cross-Modal Experiments**:
   - Consider testing MoT on more challenging cross-modal tasks (e.g., reasoning-heavy VQA or multi-turn dialogue tasks) to confirm that global self-attention alone is sufficient or to see if specialized cross-modal fusions improve performance.

2. **Implementation Details for Production**:
   - Provide a deeper discussion on dealing with variable sequence, dynamic batch sizes, and latency constraints.
   - Comment on the cached indexing or potential specialized GEMM kernels to optimize the overhead from modality-based partitioning.

3. **Inference-Time Profiling**:
   - Expand on how MoT’s multi-modal decoupling impacts inference speed under real-world conditions (e.g., single-stream vs. batch inference, mixed-modality requests).

**Strengths And Weaknesses:**

**Strengths**
1. **Efficient Sparse Architecture**: Decoupling transformer components by modality is straightforward yet highly effective. The global self-attention mechanism still supports cross-modal interactions.
2. **Extensive Experiments**: The authors validate MoT across multiple settings (Chameleon, Chameleon+Speech, Transfusion) and multiple model scales, offering strong evidence of consistency and scalability.
3. **Real-World Throughput Gains**: Beyond theoretical FLOP savings, the paper includes system profiling that shows significant wall-clock time reductions compared to dense baselines.
4. **Ablations & Analysis**: Clear ablations (e.g., which transformer components are untied) and results on combining MoT with MoE give additional insights into architectural choices.

**Weaknesses**
1. **Cross-Modal Reasoning**: While global self-attention in MoT naturally supports cross-modal interaction, it would be interesting to see if more complex cross-modal fusion layers or specialized heads would further improve tasks requiring deeper reasoning (e.g., multi-step visual QA).
2. **Production Deployment**: The paper’s system analysis is strong, but real-world deployments can face variable batch sizes and latency-sensitive requirements. More details on how MoT handles highly variable or multi-modal streaming inputs in production would enhance practical utility.
3. **Long-Range Cross-Modal Tasks**: The paper demonstrates standard benchmarks and tasks like image generation and captioning. It might be valuable to consider long-range tasks (e.g., audio-visual alignment or multi-step referencing in text+image+speech).

---

### Decision · Action_Editor_iQEX · 2025-03-09

**Recommendation:** Accept with minor revision

**Comment:**

After the author rebuttal, all reviewers are supportive and recommend acceptance. However, the authors have not yet incorporated the requested changes into their paper.

**Audience:**

This paper is highly relevant and will appeal to a broad TMLR audience.

**Claims And Evidence:**

This paper presents Mixture-of-Transformers (MoT), a sparse multi-modal transformer that lowers the computational cost of training large multi-modal language models. MoT partitions non-embedding parameters by modality while preserving global self-attention, enabling efficient processing of text, images, and speech. Experiments demonstrate that MoT matches dense baselines while significantly reducing FLOPs. In the Transfusion setting, it performs on par with or better than dense models using fewer resources. System profiling further highlights MoT’s efficiency, cutting wall-clock training time while maintaining output quality. MoT is an excellent addition to Chameleon and Transfusion.